# Evaluation of the CAMS global atmospheric trace gas reanalysis 2003-2016 using aircraft campaign observations

Yuting Wang[1,a,*], Yong-Feng Ma[2,*], Henk Eskes[3], Antje Inness[4], Johannes Flemming[4], Guy P. Brasseur[1,5]

[1]Max Planck Institute for Meteorology, Hamburg, 20146, Germany
[2]Institute of Geophysics, Faculty of Physics, University of Warsaw, Warsaw, Poland
[3]Royal Netherlands Meteorological Institute, De Bilt, The Netherlands
[4]ECMWF, Shinfield Park, Reading, RG2 9AX, UK
[5]National Center for Atmospheric Research, Boulder, CO, USA
[a]now at Department of Civil and Environmental Engineering, the Hong Kong Polytechnic University, Hung Hom, Kowloon, Hong Kong
*These authors contributed equally to this work

*Correspondence to: Guy Brasseur (guy.brasseur@mpimet.mpg.de)*

**Abstract.** The Copernicus Atmosphere Monitoring Service (CAMS) operated by the European Centre for Medium Range Weather Forecasts (ECMWF) has produced a global reanalysis of aerosol and reactive gases (called CAMSRA) for the period 2003-2016. Space observations of ozone, carbon monoxide, $NO_2$ and aerosol optical depth are assimilated by a 4-D Var method in the 60-layer ECMWF global atmospheric model, which for the reanalysis is operated at a horizontal resolution of about 80 km. As a contribution to the evaluation of the reanalysis, we compare atmospheric concentrations of different reactive species provided by the CAMS reanalysis with independent observational data gathered by airborne instrumentation during the field campaigns INTEX-A, INTEX-B, NEAQS-ITCT, ITOP, AMMA, ARCTAS, VOCALS, YAK-AEROSIB, HIPPO and KORUS-AQ. We show that the reanalysis reproduces rather successfully the observed concentrations of chemical species that are assimilated in the system including $O_3$ and CO with the biases generally less than 20 %, but generally underestimate the concentrations of the primary hydrocarbons and secondary organic species. In some cases, large discrepancies also exist for fast-reacting radicals such as OH and $HO_2$.

## 1. Introduction

Global reanalyses of the chemical composition of the atmosphere are intended to provide a detailed and realistic view of the three-dimensional distribution and evolution of the concentrations of the chemical species over a period of several years. Information provided by advanced models in which different observational data are assimilated, is provided at rather high spatial and temporal resolutions (typically 80-110 km and 3-6 hours, respectively). The Copernicus Atmosphere Monitoring Service (http://atmosphere.copernicus.eu, CAMS), operated by the European Centre for Medium Range Weather Forecasts (ECMWF) on behalf of the European commission, is currently producing a new global reanalysis of aerosols and reactive trace gases (referred to as CAMSRA). The current released reanalysis of aerosols and reactive gases covers the period 2003-2016

(Inness et al., 2019; Wagner et al., 2019), and has recently been extended to 2017 and 2018 (Christophe et al., 2019), and this reanalysis run will be continued close to real time. ECMWF has produced several other Atmospheric Composition (AC) reanalyses. The earlier Monitoring Atmospheric Composition and Climate (MACC) project produced the MACC reanalysis (MACCRA) for the period of 2003-2012 (Inness et al., 2013; Stein et al., 2012). The CAMS interim reanalysis (CIRA) is a test product implemented after the retirement of the coupled Integrated Forecast System (IFS-MOZART; Flemming et al., 2009) and its replacement by the IFS with on-line integrated chemistry and aerosol schemes ((Flemming et al., 2015). The CIRA is available from 2003 to 2018 (Flemming et al., 2017). The CAMSRA is built on the experience gained during the production of these previous two versions of the reanalysis, MACCRA and CIRA.

The validation of the CAMSRA is routinely performed by the CAMS validation team through the CAMS-84 contract co-ordinated by KNMI (Christophe et al., 2019; Eskes et al., 2015, 2018). The validation uses various measurements, including satellite observations, ground-based remote sensing and in-situ measurements, ozone soundings and commercial aircraft measurements, to assess the performance of the model versions and the reanalysis. The validation results for CAMSRA 2003-2016 using these operational measurements are shown by Eskes et al. (2018) and Wagner et al. (2019). The purpose of our paper is to report on the validation of the CAMSRA by using aircraft measurements performed during past field campaigns in different parts of the world.

In contrast to the long-term operational monitoring, aircraft campaigns are designed to address specific scientific questions, and perform intensive measurements in a specific region during a limited period of time. Aircraft campaigns are therefore valuable supplements to evaluate the models and in particular the reanalyses. Another advantage of intensive campaigns is that they provide the opportunity to measure the concentrations of the chemical species that are not operationally monitored. The observations of these additional species can be used to better investigate the performance of the models and in particular their ability to represent some complex physical and chemical processes (Emmons et al., 2000).

Ozone ($O_3$) and carbon monoxide (CO) are two of the main chemical species that are simulated in the three reanalyses (MACCRA, CIRA, and CAMSRA). Satellite measurements of these species are assimilated in these three reanalyses resulting in analysed concentrations forced by observations (Inness et al., 2019), but with constraints that differ from species to species: these are strong in the case of CO and stratospheric ozone, but weaker in the case of tropospheric ozone and $NO_2$ (due to the short lifetime of this last species; Inness et al., 2015). The weaker constraint in tropospheric ozone also results from the fact that observed ozone amount in this lower region of the atmosphere is provided by the difference between total and stratospheric ozone columns. Knowledge of the distribution of ozone and CO is key for understanding the role of the chemical and transport processes in the atmosphere. Ozone is a key indicator of photochemical pollution. This molecule is produced in the atmosphere by the reactions between nitrogen oxides ($NO_X = NO + NO_2$), CO, and volatile organic compounds (VOCs) in the presence of sunlight. Hydrogen radicals ($HO_X = OH + HO_2$) play an important role in this nonlinear process (Jacob, 2000; Lelieveld and

Dentener, 2000). The photolysis of ozone followed by the reaction of the resulting electronically excited oxygen atom with water vapor (H2O) represents the main sink of tropospheric $O_3$ (Sheel et al., 2016). Carbon monoxide, either emitted at the surface by incomplete combustion of fossil fuels and biomass burning or produced in the atmosphere as a result of the oxidation of hydrocarbons (Khalil and Rasmussen 1984; 1990; Fortems-Cheiney et al., 2011), is destroyed mainly by reaction with the
70 OH radical (Pressman and Warneck, 1970). In this paper, we mainly evaluate the concentration of $O_3$ and CO produced by all the three reanalyses, by comparing them with atmospheric observations made along flight tracks during past field campaigns (see Table 2 below). These comparisons are performed in different regions of the world.

Other chemical species ($NO_X$, $HO_X$, organics) produced by the CAMSRA are also evaluated at selected locations. The
75 hydrocarbons considered are ethene ($C_2H_4$), ethane ($C_2H_6$) and propane ($C_3H_8$). Secondary organic compounds, including methanol ($CH_3OH$), acetone ($CH_3COCH_3$), ethanol ($C_2H_5OH$), and methyl hydroperoxide ($CH_3OOH$), are the products of hydrocarbons and CO oxidation. Peroxyacetyl nitrate (PAN) and nitric acid ($HNO_3$) are produced by photochemical reactions involving $NO_X$ (Emmons et al., 2000). Hydrogen peroxide ($H_2O_2$) represents a major tropospheric sink for $HO_X$ radicals. Formaldehyde (HCHO) is mainly produced by the oxidation of hydrocarbons, but also directly emitted to the atmosphere from
80 industry sources; it has a substantial impact on the $HO_X$ concentration. By comparing these species, the underlying processes in the model can be further evaluated.

## 2. Model description

Three versions of the global reanalysis are evaluated by conducting a comparison of the calculated fields with available measurements made from aircraft during selected field campaigns. Some of the key setups of these three reanalyses are listed
in Table 1. The chemical schemes adopted for the reanalysis models are the MOZART-3 mechanism (Kinnison et al., 2007) in the case of MACCRA, and a modified version of the Carbon Bond 2005 chemistry mechanism (Huijnen et al., 2010) in the case of CIRA and CAMSRA. Surface boundary conditions for the reactive gases are generally expressed as emissions and deposition, and account for biogenic, anthropogenic and pyrogenic effects. Methane, carbon monoxide and OH are calculated interactively with, in the case of methane specified surface concentrations. More details can be found in Inness et al. (2019).
MACCRA covers the period 2003 to 2012, while CIRA and CAMSRA provide three-dimensional global fields from 2003 to 2016. Thus, in our analysis, the campaigns that took place after 2012 are excluded when compared to MACCRA. The model resolution for MACCRA and CAMSRA is equal to 80 km, while it is equal 110 km in the case of CIRA. All three reanalyses are made with a 60 vertical levels model and extend from the surface to the altitude pressure of 0.1 hPa. Each reanalysis provides two different outputs: an analysis and a 0-24 h forecast. These two fields were compared in the case of CAMSRA,
and they appear to be very similar (not shown here). The time resolution for the analysis fields is six hours for MACCRA and CIRA and three hours for CAMSRA. For the forecast fields, the time resolution is three hours for all the reanalysis versions. To use same time resolution for the three reanalyses, the forecast fields are used in this present study. The satellite datasets

that are assimilated in CAMSRA are summarized in Table 2. $O_3$, CO, and $NO_2$ are assimilated in CAMSRA, and each species is assimilated independently from the others (Inness et al., 2019). $O_3$ total column, stratospheric partial column, and profile retrievals from several satellite are used to constrain mainly the stratospheric $O_3$. As indicated above, the tropospheric forcing is weaker because the information is provided by the residual between total and stratospheric columns (Inness et al., 2015). The MOPITT total column CO retrievals are assimilated in CAMSRA, and the retrievals are mostly sensitive in the mid- and upper troposphere (Deeter et al., 2013), leading to the strongest constrain in that region. MOPITT data used in the CAMS assimilation cover only the latitudes between 65 $^0$N and 65 $^0$S, so that the constraints are weak at high latitudes. For $NO_2$, the impact of the assimilation is small because the lifetime of $NO_2$ is short (Inness et al., 2015). An additional control run for CAMSRA without data assimilation is also evaluated to separate the impact of the assimilation from the other model-related factors.

When comparing the concentrations calculated in the reanalyses with the campaign data, the 4D model grid points (space and time) that are considered are those that are closest to the measurement locations (latitude, longitude, and pressure layer) and times.

## 3. Aircraft measurements

Several aircraft campaigns are used to validate the three CAMSRA presented above. These campaigns are briefly described below, and in Table 2.

INTEX-A (Intercontinental Chemical Transport Experiment – North America Phase A) was an integrated atmospheric field experiment performed over the east coast of the United States organized by NASA during July and August in 2004 (Singh et al., 2006). It has contributed to a large ICARTT program (International Consortium for Atmospheric Research on Transport and Transformation; Fehsenfeld et al., 2006). During this campaign, the chemical species were measured by different instruments on board of a DC-8 air plane. The measurement methodology of the trace gases can be found in Singh et al. (2006).

NEAQS-ITCT (New England Air Quality Study – Intercontinental Transport and Chemical Transformation) was the NOAA component to the ICARTT program. The instruments were setup on a WP-3D aircraft, and the details can be found by Fehsenfeld et al. (2006).

ITOP (Intercontinental Transport of Pollution) was the European (U. K., Germany, and France) contribution to ICARTT project. In the present study, we collect the measurements made on board of the UK FAAM BAE-146 aircraft. The instrument information is provided by Cook et al. (2007).

INTEX-B (Intercontinental Chemical Transport Experiment – Phase B) was the second phase of the INTEX-NA experiment led by NASA. In March of 2006, INTEX-B operated in support of the multi-agency MIRAGE/MILAGRO (The Megacity Initiative: Local and Global Research Observations; Molina et al., 2010) project with a focus on observations in and around Mexico City. In its second phase, INTEX-B focused on the east coast of U. S. and on the Pacific Ocean during the spring of 2006 (Singh et al., 2009). The NCAR component of MILAGRO was MIRAGE-Mex (Megacities Impact on Regional and

Global Environment), and NCAR also contributed to INTEX-B. The NASA measurement platform was the DC-8 research aircraft. The measurement approaches for the selected species were the same as those adopted for INTEX-A. The NCAR measurements were made from the NSF/NCAR C-130 airplane. The measurement method is described by Singh et al. (2009).

AMMA (African Monsoon Multidisciplinary Analysis) was an international project to improve our knowledge and

understanding of the West African monsoon (Lebel et al., 2010). Measurements to investigate the chemical composition of the middle and upper troposphere in West Africa during July to August 2006 campaign were performed by the U. K. FAAM BAE-146 aircraft, and the details are described by Saunois et al. (2009).

ARCTAS (Arctic Research of the Composition of the Troposphere from Aircraft and Satellites) was conducted during April

and July 2008 by NASA (Jacob et al., 2010). ARCTAS was part of the international POLARCAT program during the 2007-2008 International Polar Year (IPY). In the present study, we use the measurements made on board of NASA DC-8 research aircraft. The species measured during ARCTAS were the same as during INTEX-A.

VOCALS (VAMOS Ocean-Cloud-Atmosphere-Land Study) was an international program that is part of the CLIVAR

VAMOS (Variability of the American Monsoon Systems) project. The VOCALS experiment was conducted from 15 October to 15 November 2008 in the Southeast Pacific region (Allen et al., 2011). The NSF C-130 aircraft was used during the campaign.

YAK-AEROSIB (Airborne Extensive Regional Observations in Siberia) was a bilateral cooperation activity coordinated by

researchers from LSCE in France and IAO in Russia. It aims to establish systematic airborne observations of the atmospheric composition over Siberia. In the present study, we used the $O_3$ and CO measurements during 2006 - 2008 and in 2014. The program used a Tupolev Tu-134 aircraft. The detailed measurement techniques can be found in Paris et al. (2008; 2010).

HIPPO (HIAPER Pole-to-Pole Observations), supported by NSF and operated by NCAR, used the NSF/NCAR G-V aircraft.

During five missions from 2009 to 2011in different seasons, a large number of chemical species were observed between the Arctic and the Antarctic over the Pacific Ocean. The details can be found in Wofsy et al. (2012).

KORUS-AQ (Korea-US Air Quality Study) was a joint Korea and U. S. campaign that took place in South Korea from April to June 2016. The U. S. contribution was led by NASA, and the aircraft platform was the NASA DC-8. The species were measured as during the INTEX-A campaign. A further description of this field campaign can be found in the KORUS-AQ White Paper (https://espo.nasa.gov/korus-aq/content/KORUS-AQ_Science_Overview_0, last access: 10 July 2019).

Since the goal of the present study is to evaluate the different ECMWF reanalyses by comparing the calculated fields with observations conducted during different campaigns and using different instruments, it is important to state that the measurements of the major species are comparable. The different instruments deployed during these campaigns were all carefully calibrated, and in the case of ozone and carbon monoxide, for example, the quoted uncertainties in the measurements is 3-5 ppb and 2-5 ppb respectively, depending on the instrument. When, for a given campaign, more than one instrument was used, the quantitative values were comparable, and were averaged before being used in our analysis. This was the case, for example, for the HIPPO campaign during which ozone was measured by two different instruments and carbon monoxide by three instruments.

Information of aircraft campaigns is summarized in Table 3. The flight tracks are shown in Figure 1.

## 4. Evaluation of spatial distributions of chemical species

In the present Section, we first evaluate the CAMSRA by comparing the calculated (reanalysed) and observed concentrations of ozone, carbon monoxide and other chemical species in different regions of the world during the selected field campaigns. Carbon monoxide and ozone were measured in all the field campaigns considered in the present study. Data are available in both hemispheres, but principally in the regions of North America, eastern Asia, Australia and across the Pacific Ocean. In the case of nitrogen oxides, hydroxyl and peroxyl radicals and formaldehyde, only the measurements provided in North America, the northern Pacific and eastern Asia are considered here.

### 4.1 Ozone

For the spatial evaluation, all the aircraft measurements and the extracted model data points are combined regardless the time of the measurement; observations and models are separated into three altitude layers: the low troposphere layer (0-3 km), the middle troposphere layer (3-9 km), and the upper troposphere/lower stratospheric layer (9-14 km).

The comparison of $O_3$ between the observation and the reanalyses is shown in Figure 2, 3, 4. The tropospheric ozone concentration is higher in the northern hemisphere than the southern hemisphere because of higher anthropogenic emissions of ozone precursors (air pollution). In the 9-14 km layer, the polar ozone concentrations are very high because the height of the tropopause in that region is lower than at lower latitudes, and as a result, the aircraft penetrated in the ozone-rich

stratosphere. The comparison between the aircraft observations and the reanalysis values from MACCRA is generally good. In the low troposphere, the biases of the averaged grids are mostly within 20 %. MACCRA underestimates the $O_3$ concentrations in the Arctic region and in the southern hemisphere; while it overestimates the $O_3$ concentrations in the northern low and mid-latitudes, especially over the western Pacific Ocean (over 50 %), the eastern coast of U. S. and the North Atlantic (about 40 %). The biases of MACCRA in the middle troposphere are smaller than those in the low troposphere. The positive biases in the lower layer become smaller with increasing altitude everywhere except in the Arctic, where the negative biases turn to positive values. In the upper troposphere, the agreement is worse than in the lower layers. The biases are mostly positive over the Pacific Ocean and negative in North America.

The agreement of CIRA with the aircraft measurements is similar to the agreement of MACCRA when using the same measurements before 2013. In the lowest layer, however, the mean bias of CIRA is slight smaller than that of MACCRA. The CIRA reanalysis overestimates the observation in the middle of the Pacific Ocean and northwest of the Atlantic Ocean, which is similar to the values derived from MACCRA, but with smaller biases; CIRA underestimates ozone concentrations in the rest of the region with biases of less than 20 %. Above the Pacific Ocean, the positive bias, which is small in the lower layers of the atmosphere, increases with height and becomes substantial in the upper troposphere. The patterns of the biases in the CIRA reanalysis in the upper troposphere are similar to those in the middle troposphere layer, but with larger values.

In the low troposphere, CAMSRA generally overestimates the $O_3$ concentration relative to the observation, which is different from the MACCRA and CIRA cases. The biases of CAMSRA are usually less than 15 %, and the relative larger biases are found in the tropics and Arctic, where the reanalysis overestimates the measurements by about 30 %. In the free troposphere, the biases of the reanalysis become larger than in the low troposphere, especially over the tropical ocean, while the differences are smaller in the western African region. For the comparison above 9 km, the positive biases over the Pacific Ocean are even larger and reach 50 %.

The mean bias of the CAMS control simulation (model run performed without assimilation of observed data) is similar to the bias associated with CAMSRA, but the patterns are different. The bias of CAMSRA is more uniform over the globe, which shows that data assimilation improves the global distribution of the $O_3$ concentration. In the low troposphere, the bias of the control run is of the order of 15 %. The control run underestimates the measurements in the west coast of U.S. and in the south of the Pacific Ocean, where the ozone concentrations provided by the CAMSRA are higher than the observation. In the polar free troposphere, the control simulation provides concentration values that are lower than suggested by the observations with a bias of about 20 %; in contrast to this, in the tropical region, the control simulation overestimates ozone, which is similar to the corresponding estimates by CAMSRA. In the upper layer, the bias pattern is similar to that in the free troposphere, but the bias values are larger.

Overall, for ozone, the level of agreement between the observations and the three reanalyses, and between the observations and the control run are similar, but the biases associated with CAMSRA are more uniform in space. A linear regression was performed between all observed ozone data points and ozone concentrations extracted from the three reanalyses and from the control simulation. Table 4 lists the corresponding linear regression parameters. Fewer data are available when considering MACCRA because MACCRA includes information only until year 2012. To more directly compare with MACCRA, the regression parameters for the other models runs before 2013 are also given in the table. The correlations of all the three reanalysis cases are high with squared correlation coefficients larger than 0.9. The highest correlation is achieved with CAMSRA. The squared correlation coefficient $R^2$ derived for the control simulation (0.89) is not substantially smaller than in the three cases with assimilation (0.93). This suggests that the CAMS model in its control mode has good predictive capability, but that, as expected, data assimilation slightly improves the calculated ozone fields. To exclude the contribution of stratospheric ozone values in the statistical analysis, the stratospheric data were filtered out, and the statistical parameters recalculated. The squared correlation coefficients decreased from about 0.9-0.95 to about 0.6-0.7.

## 4.2 Carbon monoxide

The comparison of carbon monoxide between the observation and the reanalyses is shown in Figures 5, 6 and 7. MACCRA underestimates the CO concentrations in the Arctic region and Canada (about 30 %), west Africa (about 20 %), and the Southern Ocean (about 10 %). It overestimates the concentrations in the other regions covered by the campaigns with most of the biases within 15 %. In the middle troposphere, the bias pattern is similar to that of the low troposphere, but the biases are smaller than in the lowest layer, especially in the Arctic. In the upper layer, often located in the stratosphere, the biases become larger at high latitudes (positive in the Arctic and negative in the Southern Ocean), with the biases larger than 50 %. In this layer, the patterns of the biases over the Pacific Ocean are different than in the lower layers.

CIRA agrees better with the observations than MACCRA. In the low troposphere, the biases are smaller than those derived with MACCRA, and the large negative biases in the Arctic found in MACCRA disappear with CIRA. The mean bias of CIRA is only of the order of 10 %. CIRA underestimates the CO observation in the region of the northern Pacific Ocean, but MACCRA overestimates the concentrations there. In the middle troposphere, CIRA underestimates CO in most regions in the northern hemisphere, while it overestimates CO in the southern hemisphere. In the upper layer, the biases of CIRA are also large at high latitudes, but the biases are positive in the both polar regions.

The agreement between the CO measurements and the CAMSRA is generally good with biases generally smaller than 15 %. In the low and middle troposphere, the CAMSRA behaves similarly to CIRA; however, in the upper layer, the biases are different. The biases in CAMSRA become smaller in the polar region. CAMSRA underestimates CO concentrations in most regions of the low and mid latitudes with biases less than 20 %.

The bias between the control run and the CO observations is larger than for the CAMSRA. The bias pattern of the control run in the lowest layer is similar to that of CAMSRA, but the positive biases in the southern hemisphere are larger (about 30 %). In the free troposphere, the control run underestimates the CO concentration at latitudes north of 40 °N, similar to the CAMSRA, but overestimates the CO elsewhere. The positive model biases in the southern hemisphere and tropics are
efficiently removed by the assimilation of CAMSRA. In the upper layer, the biases are positive in most regions except west of North America. The biases are large in the polar stratosphere, where they reach about 50 %.

When confronted with CO data collected by airborne instrumentation, all three reanalyses provide good results in the low and middle troposphere; however, the two early reanalyses are not successful when considering the field observations made in the
polar region, specifically in the upper troposphere/lower stratosphere. The situation is improved with the new CAMSRA reanalysis. The control simulations performed without assimilation overestimate the CO concentration in the southern hemisphere. The linear regression parameters of CO are shown in Table 5. For all the data points, the correlations are weak due to the extreme values appearing in localized pollution plumes, not captured by coarse resolution global models. After filtering out these extreme values (values larger than 300 ppb), the correlations of CO between the observations and models
improve substantially. The correlation calculated using CAMSRA is the highest with a correlation coefficient of 0.71 and a slope of 0.78. The mean bias of CAMSRA reduces with the assimilation resulting from the correction of the positive bias in the Southern Hemisphere.

## 4.3 Other chemical species

Spatial distributions of nitrogen oxides ($NO_X$ = $NO$ + $NO_2$), hydroxyl radical (OH), hydroperoxyl radical ($HO_2$), and
formaldehyde (HCHO) for CAMSRA in the northern hemisphere are provided in the appendix. The CAMSRA reanalysis values are compared with observations from aircraft for three different layers of the atmosphere. Because the measurements of $NO_X$, OH, $HO_2$, and HCHO used in the work are only in North America, Arctic, and Korea, so the analysis below are for these regions.

In the case of $NO_X$, the CAMSRA reanalysis underestimates the values measured in the middle and upper troposphere, but overestimates the observed values in the lowest layer. There are several possible reasons: (1) the model overestimates the effect of regional pollution sources; (2) the model underestimates the local productions (e.g. lightning); (3) the model underestimates the convective transport; (4) the model underestimates the lifetime of the surface emissions. We also compared the $NO_X$ fields produced by CAMSRA and the control run in order to assess the benefit of $NO_2$ assimilation. Both fields are
very similar, which suggests that the assimilation does not significantly improve the reanalysis of $NO_X$. This is explained by the fact that $NO_2$ has a short lifetime. Most of the impact of the data assimilation is therefore lost between analysis cycles (Inness et al., 2015). In the case of HCHO, the reanalysis underestimates the observed concentrations at all levels. The negative biases in the low troposphere are between 20-40 %, while those in the higher levels are about 50 %. In the case of OH, the

calculated values are overestimated at mid and low latitudes, which may lead to a shorter lifetime of $NO_2$, consistent with the vertical distribution of $NO_X$ discussed above. CAMSRA underestimates OH concentrations in the Arctic region, which may be related to the overestimation of CO in that region. Finally, no clear pattern is found in the difference between model simulated values and observations of $HO_2$.

## 5. Evaluation of vertical profiles at selected locations

The CAMS reanalysis provides the global distribution of a large number of chemical species that are not directly assimilated by the CAMS system, but whose concentrations are calculated consistently with the assimilated species, ozone, carbon monoxide and nitrogen dioxide. We evaluate several key species calculated by CAMSRA at four selected locations with observations from NASA campaigns (INTEX-A in 2004, INTEX-B in 2006, and ARCTAS in 2008) that took place with the DC-8 research aircraft (Figure 8). These campaigns provide information on the atmospheric abundance of several reactive gases related to ozone and CO chemistry. The vertical profiles at the chosen locations are averaged based on ARCTAS campaign in the case of the Arctic region (measurements north to 60 °N), on the INTEX-B campaign in the case of Hawaii and Mexico and on INTEX-A in the case of the Bangor data. Since only $O_3$, CO, and $NO_2$ are assimilated in CAMSRA reanalysis, the control simulation without assimilation is shown only for $O_3$, CO, and $NO_X$. A comparison between the reanalysis and the control simulations for species other than $O_3$, CO and $NO_X$ is not shown because the differences between the two runs are very small. The vertical profiles of ozone, carbon monoxide, nitrogen oxides ($NO_X$), hydroxyl (OH) and hydroperoxyl ($HO_2$) radical, formaldehyde (HCHO), hydrogen peroxide ($H_2O_2$), nitric acid ($HNO_3$), peroxyacetyl nitrate (PAN), ethene ($C_2H_4$), ethane ($C_2H_6$), propane ($C_3H_8$), methanol ($CH_3OH$), acetone ($CH_3COCH_3$), methyl hydroperoxide ($CH_3OOH$) and ethanol ($C_2H_5OH$) are shown in Figures 9, 10, 11, and 12.

We first examine the case of the three assimilated species. In general, the profiles calculated with assimilated observations are in good agreement with the profiles observed by airborne instruments. There are some interesting points to note, however.

### 5.1 Ozone

In the case of ozone in the Arctic (Figure 9), where the vertical profile is strongly affected by stratospheric processes, the control run underestimates the $O_3$ concentration above 1 km and particularly above 6 km altitude. The assimilation brings the profile much closer to the aircraft data. The concentrations calculated by the control and the reanalysis runs in the surface layer below 1 km are almost twice as large as those derived from the observations, which may be affected by the halogen chemical removal in Arctic spring. In the free troposphere and low stratosphere, the agreement is best for CAMSRA. In Bangor (Figure 10), the control and reanalysis simulations underestimate the aircraft observations in the upper troposphere, while they overestimate the measurements near the surface.

The low-latitude ozone profiles (Figures 11 and 12) are well reproduced by the reanalysis. However, the control run tends to overestimate ozone in Mexico-City and to a lesser extend in Hawaii. In this last region, the agreement of $O_3$ between the observations and models is quite good below 7 km: the biases are positive and smaller than 10 %, which is opposite to what is found in the Arctic. The reanalysis provides slightly better results than the control run. At higher altitudes the positive biases get larger and the CAMSRA data become worse than in the control run, which is surprising since the model with assimilated ozone should be better constrained. This result may be due to the fact that the constraint on tropospheric ozone is weak and the bias correction may be distributed incorrectly in the vertical. In Mexico-City, the model represents well the ozone bulge that is detected by the airborne instruments at 2 to 3 km and is observed for most chemical species. At higher altitudes, the control model overestimates the ozone concentration; however, the bias is reduced by the CAMSRA assimilation.

## 5.2 Carbon monoxide

In the case of Arctic CO (Figure 9), the general agreement between the control and reanalysis runs and the observed profile is very good. The control run, however, slightly underestimates the CO concentration in the troposphere but overestimates it in the stratosphere. The assimilation does not change the simulation significantly as MOPITT observations with latitudes higher than 65 degrees were excluded in the CAMS assimilation. It increases the biases in troposphere CO in CAMSRA but decreases the positive biases in the stratosphere. In Bangor (Figure 10), both the control and the reanalysis simulations underestimate the observed concentrations by typically 10 ppb above 3 km altitude, but underestimate the surface concentrations. At low latitudes (Hawaii and Mexico, Figures 11 and 12), the control simulation overestimates the concentrations by about 10 ppb in the free troposphere, while CAMSRA underestimates the values observed from the DC-8 by 10 ppb. In Hawaii in the first 2 km above the surface, the control run provides concentrations that are about 10 % lower than the aircraft observation. In Mexico, the control model provides surface values that are 30 % higher than the observation. The bulge observed at 2-3 km altitude is not reproduced by the model.

## 5.3 Nitrogen oxides

In the Arctic (Figure 9) the control run underestimates the $NO_X$, especially above 8 km, i.e., in the layers strongly influenced by the injection of stratospheric air. The assimilation process does not substantially reduce the discrepancy, since the CAMS model does not include a detailed representation of stratospheric chemistry and $NO_X$ in the stratosphere is strongly underestimated because of this. In Bangor (Figure 10), the models underestimate the $NO_X$ above 2 km as in Arctic, but overestimate the $NO_X$ below 2 km. In the low latitude regions (Mexico and Hawaii, Figures 11 and 12), the calculated profiles are in rather good agreement with the observations, except below 2 km, where the influence from local air pollution is not well captured by the control and reanalysis simulations. In Hawaii, the model tends to slightly underestimate the observation. As in the Arctic, this underestimation is larger in the case of the reanalysis. In all regions except the Arctic, the models provide higher surface concentrations than suggested by the measurements.

### 5.4 Hydroxyl and hydroperoxyl radicals

In the Arctic (Figure 9), the model underestimates OH concentrations by about 0.02 ppt at all altitudes (of the order of 50 %), which may be linked to the slight overestimation of calculated stratospheric CO. In the reanalysis, the concentrations of $HO_2$ are overestimated by about 1 pptv between 4 and 8 km altitude. In Bangor (Figure 10), the reanalysis overestimates OH by about 0.2 pptv, which is coincident with the underestimation of the CO concentration at this location. The $HO_2$ concentrations are overestimated by 3-5 pptv. In Hawaii, the simulations made for the reanalysis overestimate the OH concentrations below 6 km but underestimates them above 8 km, which is consistent to the overestimation of high-altitude CO in the control run. In Mexico-city, the simulated OH concentrations are larger than the measurements below 8 km, but smaller above 8 km. The reanalysis overestimates $HO_2$ by about 4 pptv or 20 %.

### 5.5 Hydrogen peroxide

In the Arctic (Figure 9), where the calculated concentrations of $HO_2$ are too high in CAMSRA, the concentration of hydrogen peroxide is overestimated by typically a factor 2. In Bangor (Figure 10), the overestimation is of the order of 20 %. The agreement between reanalysis and observations is generally good in Hawaii (Figure 11) and Mexico-City (Figure 12), except in the lower levels of the atmosphere, where the model overestimates the concentrations.

### 5.6 Nitric acid

Nitric acid concentrations are strongly affected by wet scavenging in the troposphere and, at high latitudes, by the downward flux of stratospheric air (Murphy and Fahey, 1994; Wespes et al., 2007). The reanalysis generally underestimates the concentration of $HNO_3$ above 2 km altitude. This is the case in the Arctic (Figure 9), in Bangor (Figure 10) and in Hawaii (Figure 11). The discrepancy is particularly large in the upper levels of the Arctic, which implies that (1) scavenging of $HNO_3$ is too strong, (2) the reactive nitrogen (e.g. $NO_X$) in the stratosphere is too low due to missing stratospheric chemistry. The model accounts for the high concentrations observed in the lowest levels of the atmosphere, specifically in Mexico-city (Figure 12) and to a lesser extent in Bangor and Hawaii.

### 5.7 Peroxyacetyl nitrate (PAN)

The agreement between the calculated and observed PAN vertical profile is good in the Arctic (Figure 9), even though the concentrations are slightly underestimated between 2 and 8 km altitude. The agreement is also good in Hawaii (Figure 11) below 5 km altitude, but a discrepancy of about 50 % is found above this height. In Bangor (Figure 10), PAN concentrations are overestimated by about 25 % in the free troposphere and by as much as a factor 2 below 3 km altitude. The calculated concentrations are slightly too high in Mexico-city (Figure 12). The model shows the presence of a peak in the PAN concentration at 3 km, but the calculated concentrations values are somewhat too low.

 **5.8 Primary organic compounds: ethene ($C_2H_4$), ethane ($C_2H_6$) and propane ($C_3H_8$)**

In most cases, the model underestimates the measured concentrations of the primary hydrocarbons, which indicates that the emissions are too low. The discrepancy is substantial at all altitudes for example for $C_2H_4$ in Hawaii (Figure 11), as well as $C_3H_8$ in the Arctic (Figure 9) and in Bangor (Figure 10). Calculated $C_2H_6$ is substantially lower than suggested by the observations at all four locations. In Mexico-city (Figure 12), the model reproduces rather successfully the vertical profile of
$C_2H_4$, but underestimates $C_3H_8$ below 5 km altitude. This last compound is well represented in Hawaii in the upper troposphere, but is underestimated by the model below 7 km.

**5.9 Secondary organic compounds: formaldehyde (HCHO), methanol ($CH_3OH$), acetone ($CH_3COCH_3$), ethanol ($C_2H_5OH$) and methyl hydroperoxide ($CH_3OOH$)**

As should be expected from the underestimation by the reanalysis of the atmospheric concentration of the primary
hydrocarbons, the model also underestimates the abundance of oxygenated organic species in the troposphere. This is the case in the Arctic (Figure 9), where the abundance of formaldehyde, acetone and ethanol are underestimated by typically factors 3 to 8.  Methanol is too low by about 30 %. Large discrepancies are also found in Bangor (Figure 10) where methanol and acetone are underestimated by a factor 2 and methyl peroxide by a factor 5. In Hawaii (Figure 11), the concentration of formaldehyde is slightly underestimated in the middle and upper troposphere, but the discrepancy reaches a factor 2 at 2 km
altitude. Methanol is underestimated by 30 % but acetone and ethanol are underestimated by a factor 2.  The model is in better agreement with the observations in Mexico-City (Figure 12): this is the case for formaldehyde (except below 4 km where the calculated concentrations are a factor 3 too low), for methanol (except at the surface) and for methyl hydroperoxide except below 4 km. Ethanol is underestimated by a factor 2.

To summarize the discussion, we have qualified the degree of success of the reanalysis model versus the observational vertical profiles in the 4 regions of the world that are considered in the present study. The results, based on a subjective comparison between the vertical profiles derived from the CAMSRA and the profiles measured independently by airborne instruments, are presented in the following table for the altitudes of 6 km above the ground and at the Earth's surface, respectively. The symbols used in this table are the following: G for good agreement (bias < 10 %), O for overestimation by the reanalysis model (10 %
< bias < 40 %), and U for underestimation (-40 % < bias < -10 %). Double symbols (i.e., OO or UU) indicate from a subjective analysis that the disagreement is large (bias > 40 %).

**5.10 Concentration ratios**

In order to analyze the performance of the reanalysis and to reproduce the observed relationships between different reacting species, we present and discuss the vertical distribution of the concentration ratio between photochemically coupled chemical
compounds. In order to avoid the chemically and dynamically complex situation encountered in the boundary layer, we limit

this analysis to results (models and observations) obtained above 4 km altitude. We focus here on the NO/NO$_2$, PAN/NO$_2$, HNO$_3$/NO$_2$ and HO$_2$/OH concentration ratios (Figure 13).

We first examine for the 4 locations considered in the present study (Arctic, Bangor, Hawaii and Mexico) the NO/NO$_2$
concentration ratios derived from the aircraft observations of NO and NO$_2$, respectively, as well as the similar ratios produced by the control case (blue curves), reanalysis models (red curves), or derived from an approximative expression based on the photochemical theory of the troposphere (green curves). We note at all locations that the value derived from the reanalysis (with a detailed chemical scheme included) is in good agreement with the value derived from the simple photo-stationary expression:


$$\frac{[NO]}{[NO_2]} = \frac{J_{NO2}}{k_1\,[O_3] + k_2[HO_2\} + X}$$

where $J_{NO2}$ (about $10^{-2}$ s$^{-1}$ in the entire troposphere for a solar zenith angle of 45 degrees) represents the photolysis coefficient of NO$_2$, $k_1$ and $k_2$ are the rate constant of the reaction of NO with ozone and the hydro-peroxy radical (HO$_2$), respectively (Burkholder et al., 2015). Symbol $X$ accounts for the effects of additional conversion mechanisms of NO to NO$_2$. Note that, as
the temperature and the ozone number density decrease with height in the troposphere, the NO/NO$_2$ ratio tends to increase with altitude. In the lower stratosphere, the ratio is expected to decrease as the ozone concentration rapidly increases with height above the tropopause.

In the Arctic, the ratio derived from observations (typically equal to 1, see black curve) is about a factor 2 smaller than the
calculated ratio between 6 and 10 km altitude. In Bangor, its value (about 2 to 3) is higher than the model calculations. Perhaps the most interesting point is the substantial discrepancy between the models and the observations in the upper troposphere of the tropics (Hawaii and Mexico). One notes, for example, that the observed ratio does not increase as expected from theory and at 11 km for example, the calculated ratio, close to one when derived from the observations, reaches a value of the order of 4 or 5. Among possible causes for this discrepancy is an underestimation of the correction factor $X$ due to reactions not
considered in the models. Possible mechanisms include the reactions of NO with the methyl peroxy radical (CH$_3$O$_2$) and with BrO (Madronich, personal communication). CH$_3$O$_2$ plays a significant role in the NO to NO$_2$ conversion. The BrO radical is expected to affect the NO to NO$_2$ ratio if the BrO concentration becomes larger than 2-5 pptv. Another point to stress is the large uncertainty that results from dividing two mean concentration values to which substantial uncertainties are attached, so that the stated ratio derived from mean observations may be subject to a large error.


Figure 13 also shows the concentration ratio between PAN and NO$_2$ and between HNO$_3$ and NO$_2$. In the first case, the ratio derived from the models (control run and reanalysis) are in fair agreement with the ratios derived from the measurements of NO$_2$ and PAN concentrations. The ratio id decreasing with height in the Arctic and in Bangor, but is relatively constant with

height (typically 10-20) with some elevated values at some specific altitudes. In the case of the $HNO_2$ to $NO_2$ ratio, the differences between ratios derived from the models and the aircraft observations can be substantial. The control and reanalysis runs (blue and red curves) underestimate the ratio in the Arctic and in Bangor. The agreement is somewhat better in Hawaii and in Mexico, even though large differences exist at specific altitudes. These discrepancies can probably be explained by the role played by the heterogeneous conversion of nitrogen oxides into nitric acid, which depends on the chaotic behavior of clouds and aerosols in the troposphere. The green curve provides an estimate of the ratio derived from the following expression (assuming equilibrium) that ignores any heterogeneous conversion but is calculated using the observed values of OH:

$$\frac{[HNO_3]}{[NO_2]} = \frac{k_3[OH]}{J_{HNO3} + k_4\,[OH]}$$

Here $J_{HNO3}$ (about $6\times10^{-7}$ s) is the photolysis coefficient for nitric acid, while $k_3$ and $k_4$ are the kinetic coefficients for the reactions between $NO_2$ and OH and between $HNO_3$ and OH, respectively (Burkholder et al., 2015).

Finally, we show in Figure 13 the concentration ratio between $HO_2$ and OH, which is influenced by carbon monoxide, nitric oxide and ozone that, to a good approximation, can be expressed as

$$\frac{[HO_2]}{[OH]} = \frac{k_5[CO] + k_6[O_3]}{k_7[NO] + k_8[O_3]}$$

Here $k_5$, $k_6$ refer to the reactions of OH with carbon monoxide and ozone, respectively, and $k_7$ and $k_8$ to the reactions of $HO_2$ with nitric oxide and ozone, respectively (Burkholder et al., 2015). The ratio derived from observations (black curves) follows the vertical distribution of the ratios derived by the model (reanalysis, red curve) and calculated by the above equilibrium relation with observed values of CO, NO and ozone (green curve). The value of the ratio decreases from about $100 \pm 25$ at 4 km (all sites except the Arctic) to about 30-40 at 12 km in the tropics (Hawaii, Mexico) and to 10-20 at high latitudes (Bangor and the Arctic).

## 6. Summary

Overall, the reanalysis of assimilated tropospheric chemical species such as ozone and carbon monoxide by the CAMSRA system reproduces rather satisfactorily the observations made independently from aircraft platforms during the analyzed campaigns that took place between 2004 and 2016.

In the case of ozone, the $R^2$ coefficient is close to 0.9 and the RMSE ranges between 21 and 26 ppbv, depending on the reanalysis case that is being considered. The values of the same coefficient in the control case (no assimilation) are 0.89 and 25.4 ppbv, respectively. When only tropospheric ozone data are considered, the $R^2$ coefficient is of the order of 0.61 to 0.69 and the RMSE is close to 11 ppbv. The corresponding values for the control case are 0.67 and 10.6 ppbv, respectively. In other words, the assimilation procedure improves, but only slightly, the value of the statistical coefficients that are derived. Note that the RMSE is reduced by a factor 2 when only the tropospheric data are used, and the $R^2$ coefficient are reduced by 20-30 percent.

In the case of carbon monoxide, the $R^2$ coefficient varies from 0.2 to 0.4 depending on the adopted reanalysis, and the RMSE ranges from 55 to 67 ppbv. When plumes are removed from the observational data, the $R^2$ coefficient increases to 0.6-0.7 and the RMSE is reduced to 23-25 ppbv. These values are not substantially different from the coefficients obtained when the observations are compared with the control runs. But the assimilation brought the simulated CO concentrations to more uniform global distribution, which is a success of the reanalysis system.

The CAMSRA reproduced the vertical profiles of $O_3$ and CO quite well at selected four locations. For the species largely affected by the local plume (e.g. CO and $NO_X$), the CAMSRA underestimated the peak values. The simulation of OH and $HO_2$ in CAMSRA is generally satisfactory, but in some case the disagreement is big. The CAMSRA generally underestimated the primary hydrocarbons and the secondary organic compounds at all locations, implying the emissions are too low in the inventory used by CAMS system. It would be important in the future to improve in the reanalysis simulations the surface emissions of hydrocarbons and, if possible, assimilate organic species other than formaldehyde.

**Acknowledgements**

The present work was funded through the CAMS-84 (CAMS validation) contract, coordinated by the Royal Netherlands Meteorological Organization (KNMI, Henk Eskes). The Copernicus Atmosphere Monitoring Service (CAMS) is operated by the European Centre for Medium-Range Weather Forecasts on behalf of the European Commission as part of the Copernicus Programme. We acknowledge all the investigators who have made the measurements and made them available online. We thank Jean-Daniel Paris (Laboratoire des Sciences du Climat et de l'Environnement, Gif sur Yvette, France) and Philippe Nédélec (Laboratoire d'Aérologie, Toulouse, France) for providing the YAK-AEROSIB campaign data. The National Center for Atmospheric Research is sponsored by the US National Science Foundation.

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

**Table 1. Key setups of the three reanalyses**

| Reanalysis | MACCRA | CIRA | CAMSRA |
|---|---|---|---|
| period | 2003-2012 | 2003-2018 | 2003-present |
| Spatial resolution | 80 km | 110 km | 80 km |
| Vertical resolution | 60 levels | 60 levels | 60 levels |
| Temporal resolution | 6-hourly analysis fields 3-hourly forecast fields from 0 UTC up to 24 hours | 6-hourly analysis fields 3-hourly forecast fields from 6 and 18 UTC up to 12 hours | 3-hourly analysis fields 3-hourly forecast fields from 0 UTC up to 48 hours |
| Assimilation system | IFS Cycle 36r1 4D-Var | IFS Cycle 40r2 4D-Var (2003-2015) & IFS Cycle 41r1 4D-Var (2016-) | IFS Cycle 42r1 4D-Var |
| Chemistry module | MOZART3 (Kinnison et al., 2007) | CB05 & Cariolle ozone parameterization in stratosphere (Huijnen et al. 2010) | CB05 with updates & Cariolle ozone parameterization in stratosphere (Huijnen et al. 2010) |
| Anthropogenic emissions | MACCity (Granier et al., 2011) | MACCity & CO emission upgrade (Stein et al., 2014) | MACCity & CO emission upgrade (Stein et al., 2014) |
| Biogenic emissions | monthly mean VOC emissions by MEGAN2.1 (Guenther et al., 2006) for the year 2003 | monthly mean VOC emissions by MEGAN2.1 using MERRA reanalysis meteorology for 2003-2010; a climatology dataset of the MEGAN-MACC for 2011-2017 | monthly mean VOC emissions by MEGAN2.1 using MERRA reanalysis meteorology for the whole period |
| Biomass burning emissions | GFED (2003-2008) & GFAS v0 (2009-2012) | GFAS v1.2 | GFAS v1.2 |

**Table 2. The satellite datasets of trace gases assimilated in CAMSRA.**

| Species | $O_3$ (stratosphere) | $O_3$ (UTLS) | $O_3$ (free troposphere) | CO (free troposphere) | CO (surface/PBL) | $NO_2$ (free troposphere) |
|---|---|---|---|---|---|---|
| Satellites | MIPAS, MLS, SCIAMACHY, GOME-2A, GOME-2B, OMI, SBUV-2 | Indirectly constrained by limb and nadir sounders | Indirectly constrained by limb and nadir sounders | MOPITT | Indirectly constrained by satellite IR sounders | SCIAMACHY, OMI, GOME-2 |

*Note:* indirectly constraint means that there is no data in this layer assimilated for this species, but there is some impact coming from the residual of combining the data sets from the other layers.

**Table 3. List of used aircraft campaigns**

| Campaign | Date | Location | Species used | Webpage |
|---|---|---|---|---|
| INTEX-A | 2004.07-08 | East America | $O_3$, CO, NO, $NO_2$, OH, $HO_2$, HCHO, $H_2O_2$, $HNO_3$, PAN, $C_2H_4$, $C_2H_6$, $C_3H_8$, $CH_3OH$, $CH_3COCH_3$, $CH_3OOH$, $C_2H_5OH$ | https://www-air.larc.nasa.gov/missions/intexna/intexna.htm |
| NEAQS-ITCT | 2004.07-08 | East America | $O_3$, CO | https://www.esrl.noaa.gov/csd/projects/2004/ |
| ITOP-UK | 2004.07-08 | North Atlantic | $O_3$, CO | http://artefacts.ceda.ac.uk/badc_datadocs/itop/itop.html |
| INTEX-B | 2006.03-05 | West America | $O_3$, CO, NO, $NO_2$, OH, $HO_2$, HCHO, $H_2O_2$, $HNO_3$, PAN, $C_2H_4$, $C_2H_6$, $C_3H_8$, $CH_3OH$, $CH_3COCH_3$, $CH_3OOH$, $C_2H_5OH$ | https://www-air.larc.nasa.gov/missions/intex-b/intexb.html |
| AMMA-UK | 2006.07-08 | West Africa | $O_3$, CO | http://artefacts.ceda.ac.uk/badc_datadocs/amma/amma.html |
| ARCTAS | 2008.04-07 | North America to Arctic | $O_3$, CO, NO, $NO_2$, OH, $HO_2$, HCHO, $H_2O_2$, $HNO_3$, PAN, $C_2H_4$, $C_2H_6$, $C_3H_8$, $CH_3OH$, $CH_3COCH_3$, $CH_3OOH$ | https://www-air.larc.nasa.gov/missions/arctas/arctas.html |
| VOCALS | 2008.10-11 | Southeast Pacific | $O_3$, CO | http://data.eol.ucar.edu/project/VOCALS |
| YAK-AEROSIB | 2006-2008, 2014 | Russia | $O_3$, CO | https://yak-aerosib.lsce.ipsl.fr/doku.php |
| HIPPO | 2009-2011 | Pacific | $O_3$, CO | https://hippo.ornl.gov/data_access |
| KORUS-AQ | 2016.04-06 | Korea | $O_3$, CO | https://www-air.larc.nasa.gov/missions/korus-aq/index.html |

**Table 4. Linear regression of ozone between observations and models**

| | All data | | | | | | Troposphere data (>350 hPa) | | | | | |
| | *N* | *MB* | *MAE* | $R^2$ | slope | *RMSE* | *N* | *MB* | *MAE* | $R^2$ | slope | *RMSE* |
|---|---|---|---|---|---|---|---|---|---|---|---|---|
| MACCRA | 19522 | 0.59 | 13.01 | 0.9291 | 1.02 | 26.064 | 16009 | 0.21 | 9.13 | 0.6145 | 0.71 | 11.705 |
| CIRA | 22308 | -1.87 | 12.71 | 0.9298 | 0.94 | 22.472 | 18782 | -2.95 | 9.58 | 0.6498 | 0.67 | 11.232 |
| CIRA (2003-2012) | 19522 | -1.18 | 12.67 | 0.9341 | 0.94 | 23.225 | 16009 | -2.29 | 8.99 | 0.6111 | 0.66 | 10.823 |
| CAMSRA | 22308 | 1.92 | 11.90 | 0.9375 | 0.94 | 21.174 | 18782 | 1.01 | 8.77 | 0.6927 | 0.72 | 10.996 |
| CAMSRA (2003-2012) | 19522 | 2.49 | 11.97 | 0.9412 | 0.94 | 21.889 | 16009 | 1.55 | 8.32 | 0.6608 | 0.72 | 10.608 |
| Control | 22308 | -3.89 | 13.46 | 0.8935 | 0.84 | 25.398 | 18782 | -1.68 | 9.18 | 0.6687 | 0.66 | 10.611 |
| Control (2003-2012) | 19522 | -3.71 | 13.72 | 0.8966 | 0.85 | 26.662 | 16009 | -1.08 | 8.76 | 0.6229 | 0.63 | 10.155 |

*Note: N* is the number of points considered for the calculation of the correlation, *MB* the mean bias (expressed in ppb), *MAE* the mean absolute error (expressed in ppb), *R* the correlation coefficient and *RMSE* the root mean square error (expressed in ppb).

**Table 5. Linear regression of CO between the aircraft campaign observations and the reanalyses**

| | All data | | | | | | Data < 300 ppb | | | | | |
|---|---|---|---|---|---|---|---|---|---|---|---|---|
| | $N$ | $MB$ | $MAE$ | $R^2$ | slope | $RMSE$ | $N$ | $MB$ | $MAE$ | $R^2$ | slope | $RMSE$ |
| MACCRA | 18376 | -10.40 | 27.13 | 0.2005 | 0.30 | 54.921 | 17972 | -5.96 | 18.69 | 0.5992 | 0.63 | 23.346 |
| CIRA | 21353 | -6.55 | 28.56 | 0.3990 | 0.49 | 60.912 | 20254 | -2.48 | 17.72 | 0.6775 | 0.72 | 25.052 |
| CIRA (2003-2012) | 18376 | -4.86 | 23.71 | 0.3573 | 0.42 | 51.397 | 17894 | -1.65 | 16.25 | 0.6588 | 0.70 | 22.900 |
| CAMSRA | 21353 | -6.85 | 29.23 | 0.3559 | 0.49 | 66.706 | 20233 | -3.82 | 17.34 | 0.7061 | 0.78 | 25.284 |
| CAMSRA (2003-2012) | 18376 | -5.42 | 25.26 | 0.2716 | 0.40 | 60.387 | 17869 | -3.21 | 16.04 | 0.6863 | 0.77 | 23.553 |
| Control | 21353 | -0.11 | 31.78 | 0.3565 | 0.50 | 68.489 | 20187 | 2.45 | 20.15 | 0.6588 | 0.75 | 27.013 |
| Control (2003-2012) | 18376 | -0.25 | 27.74 | 0.2746 | 0.41 | 60.123 | 17881 | 2.03 | 19.11 | 0.6234 | 0.71 | 24.920 |

*Note: N* is the number of points considered for the calculation of the correlation, *MB* the mean bias (expressed in ppb), *MAE* the mean absolute error (expressed in ppb), *R* the correlation coefficient and *RMSE* the root mean square error (expressed in ppb).

**Table 6. Qualitative summary of the over- and under-estimation by the CAMSRA for several observed chemicals at 4 geographic locations and at two altitudes (6 km and surface)**

| | At 6 km | | | | surface | | | |
|---|---|---|---|---|---|---|---|---|
| | Arctic | Bangor | Hawaii | Mexico-City | Arctic | Bangor | Hawaii | Mexico-City |
| $O_3$ | G | G | G | O | OO | O | G | O |
| CO | G | U | G | U | G | O | U | O |
| NOx | UU | U | U | G | U | OO | O | O |
| OH | UU | O | G | G | UU | O | O | G |
| $HO_2$ | O | O | G | O | U | O | G | O |
| $H_2O_2$ | UU | U | G | G | UU | G | O | OO |
| $HNO_3$ | UU | U | UU | G | U | O | U | O |
| PAN | U | U | O | OO | O | OO | OO | OO |
| $C_2H_4$ | U | G | UU | UU | U | OO | O | G |
| $C_2H_6$ | UU | UU | U | U | UU | UU | UU | UU |
| $C_3H_8$ | UU | UU | U | U | UU | UU | UU | UU |
| HCHO | UU | UU | U | U | UU | U | G | O |
| $CH_3OH$ | UU | U | U | G | O | O | G | OO |
| $CH_3COCH_3$ | UU | UU | UU | UU | UU | G | UU | U |
| $C_2H_5OH$ | UU | UU | | UU | UU | O | | U |
| $CH_3OOH$ | | UU | U | U | | G | U | OO |

*Note: G = -10 % < bias < 10 %; O = 10 % < bias < 40 %; U = -40 % < bias < -10 %; OO = bias > 40 %; UU = bias < -40 %.*


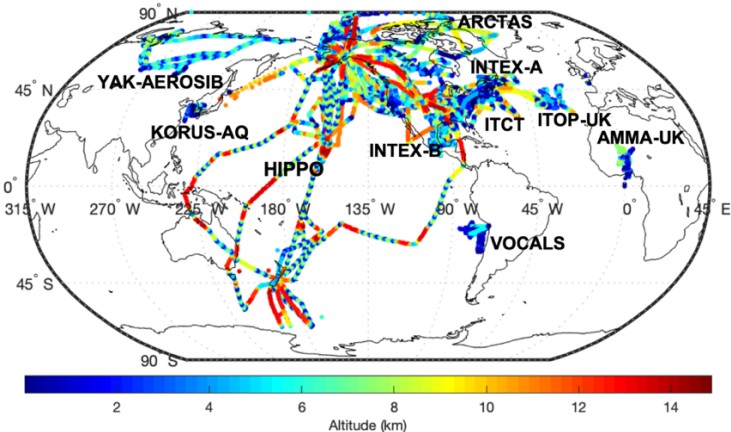

**Figure 1. Flight tracks of the campaigns with the altitude of the corresponding flight.**


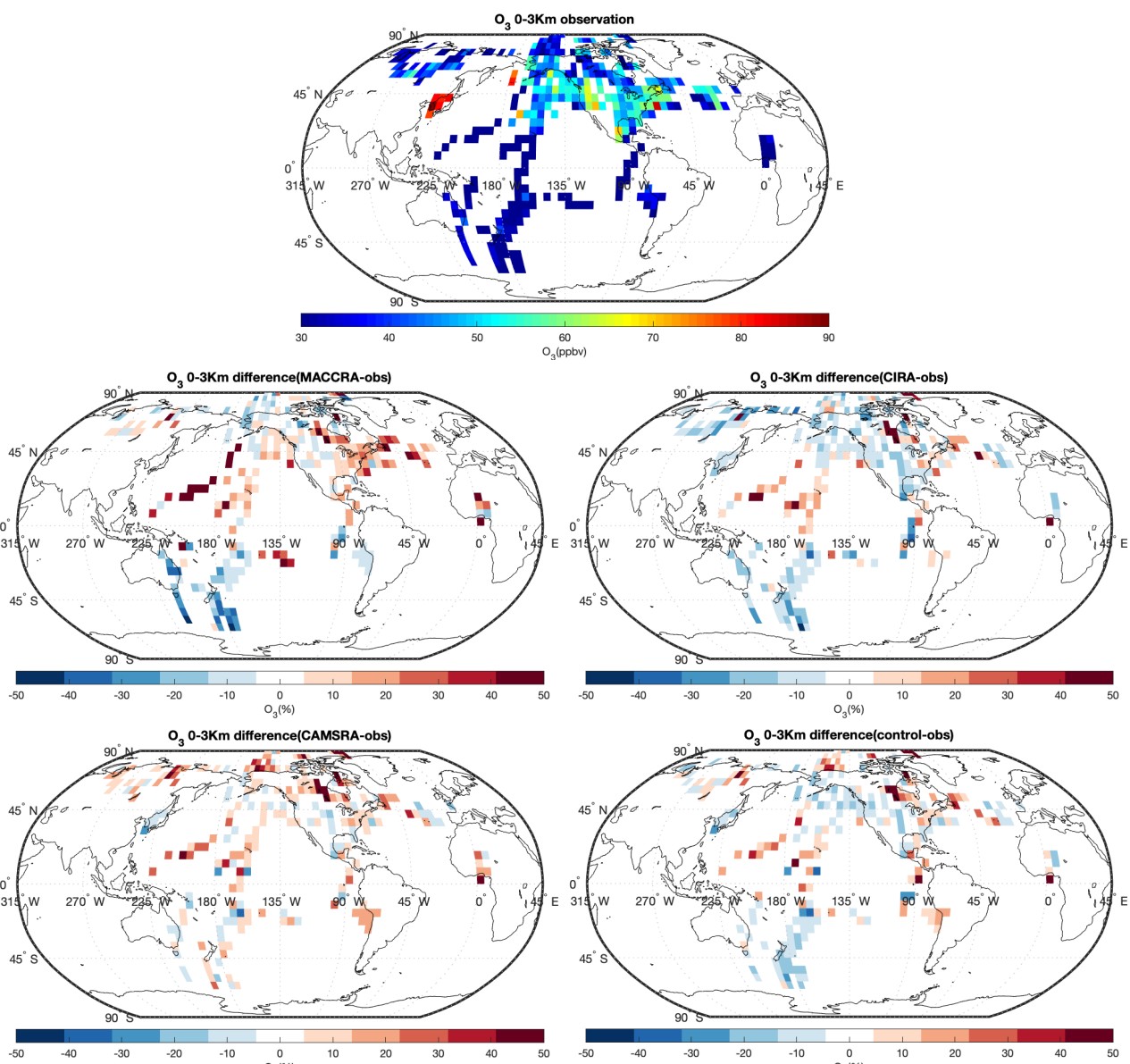


**Figure 2. Campaign observations of O₃ (first row), relative difference in % between the MACCRA and the observations (MACCRA − observation; left of second row), difference between the CIRA and the observation (CIRA − observation; right of second row), difference between the CAMSRA and the observation (CAMSRA − observation; left of third row), and the difference between the control run and the observation (control − observation; right of third row). The data are averaged to 5°×5° (latitude × longitude)**

**and to the altitude bin of 0-3 km. Note that MACCRA only includes campaigns between 2003-2012.**

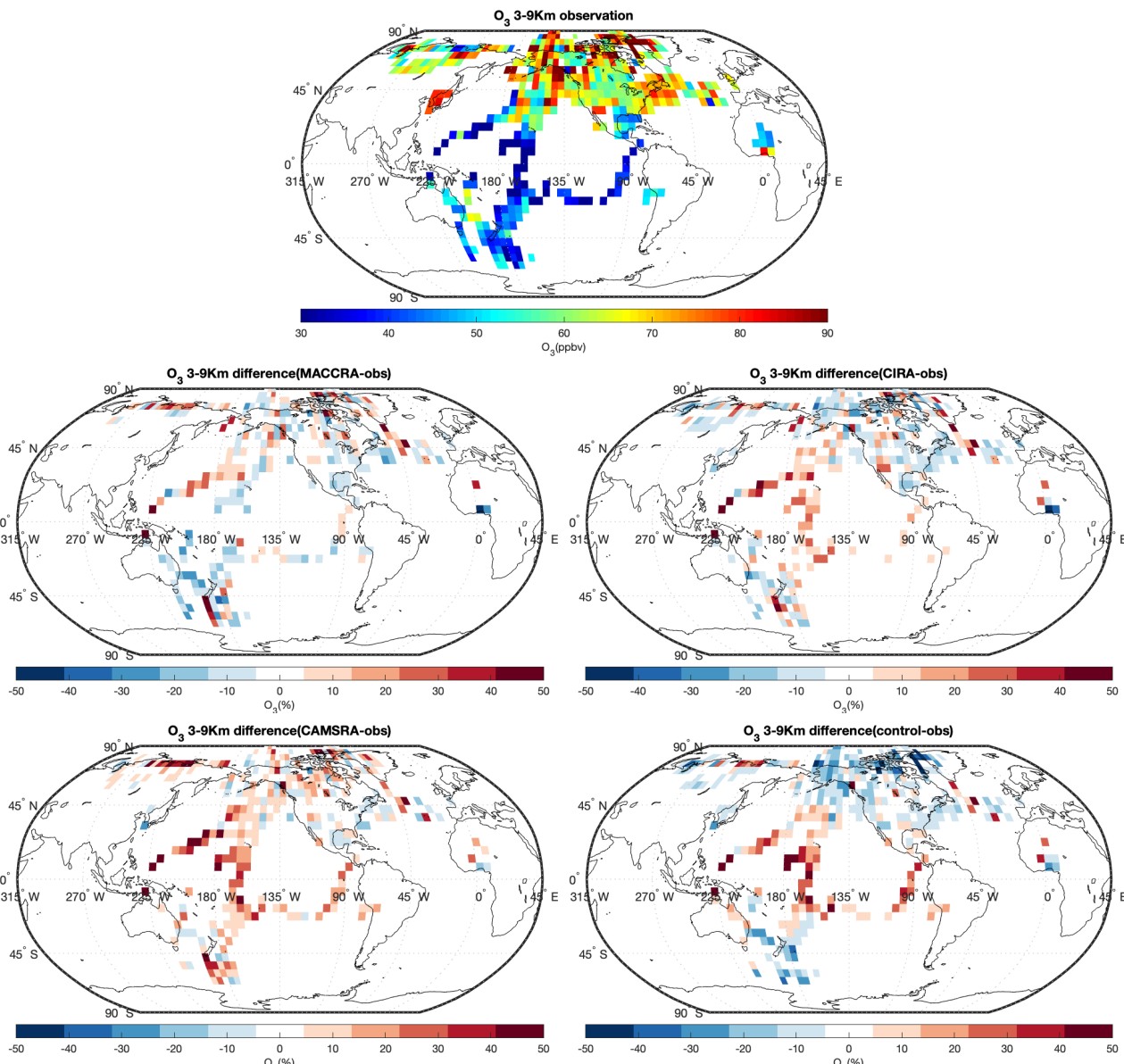

**Figure 3. Same as Figure 2, but for altitude bin of 3-9 km.**

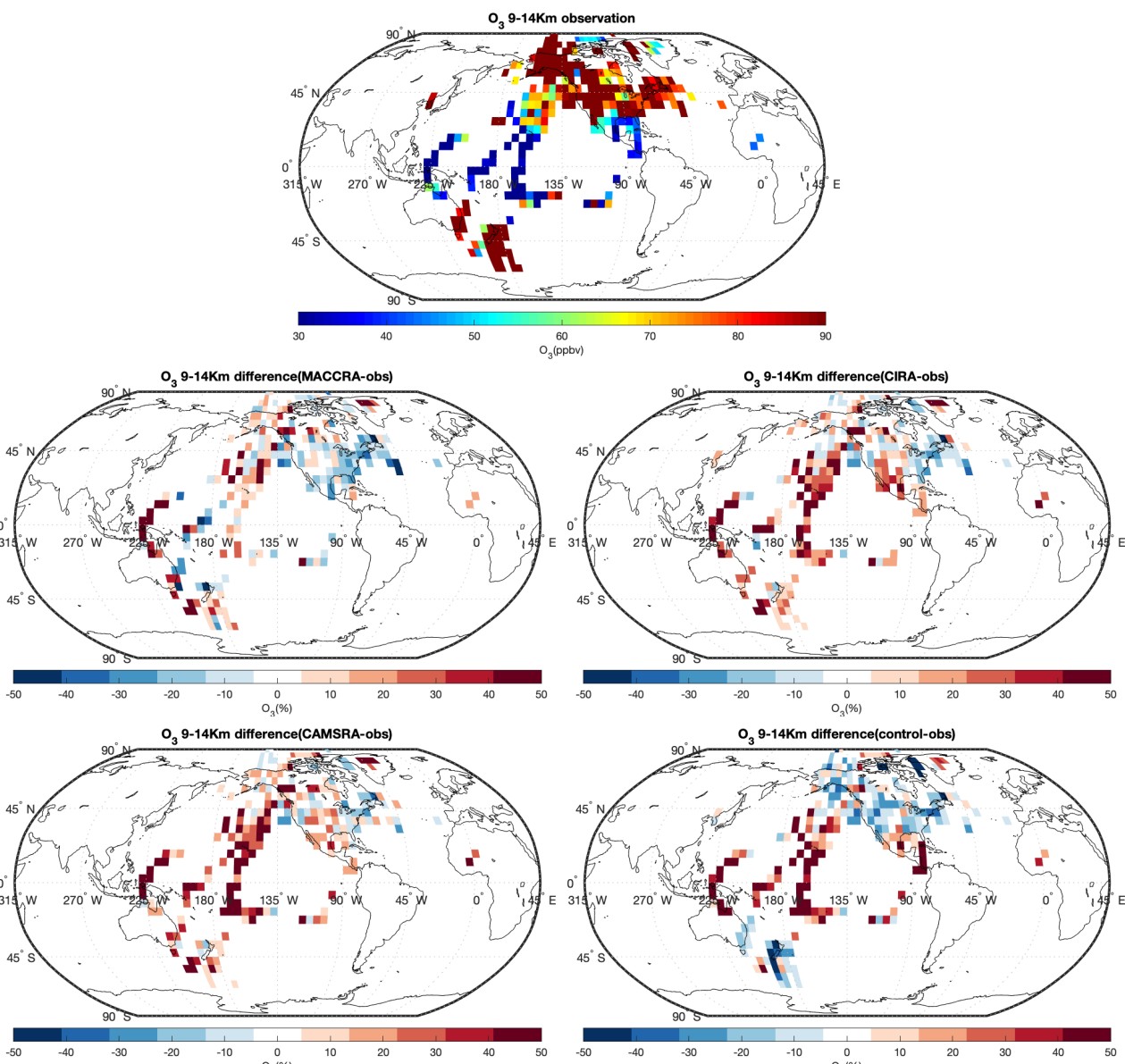

**Figure 4. Same as Figure 2, but for altitude bin of 9-14 km.**

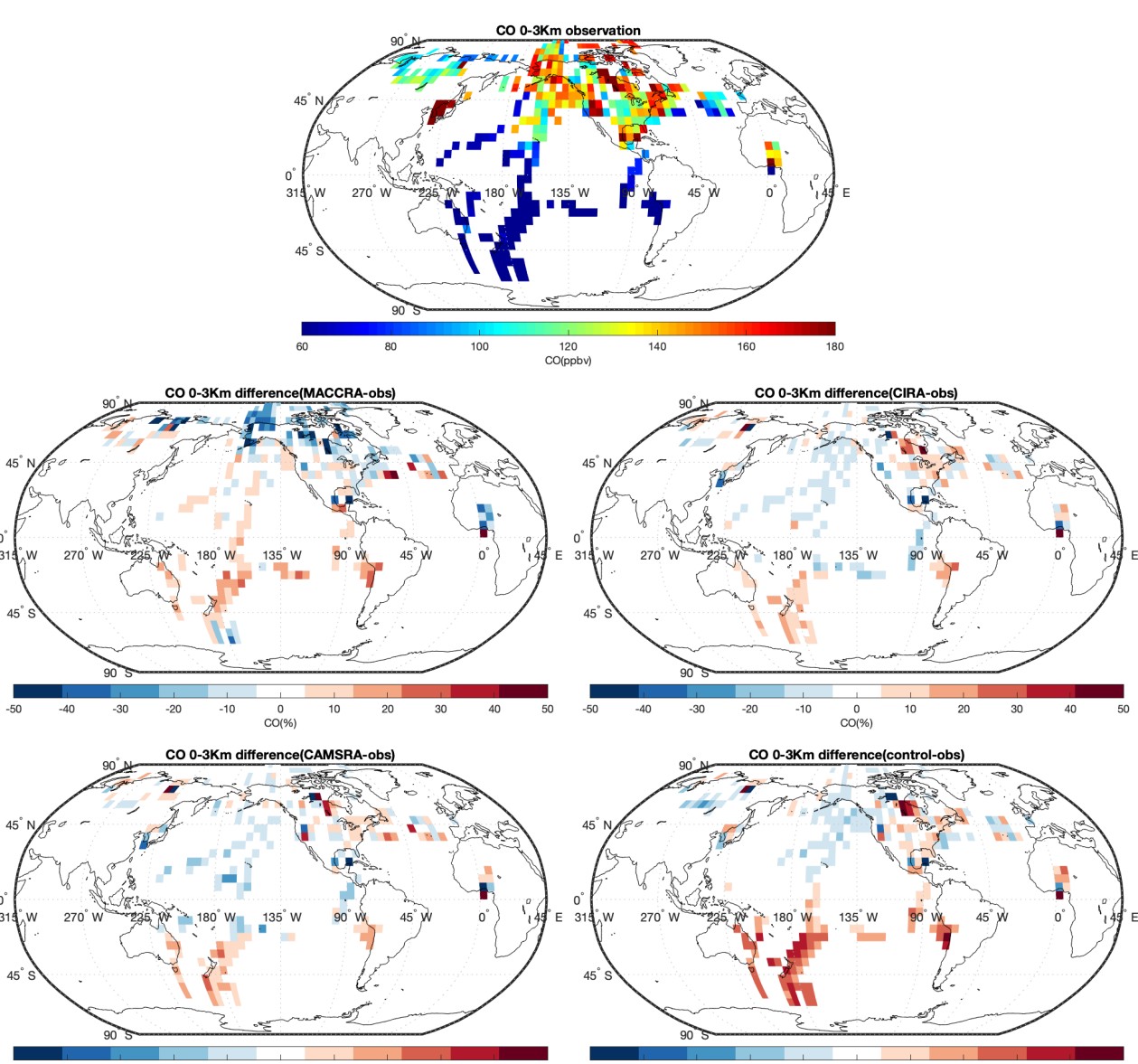

Figure 5. Campaign observations of CO (first row), relative difference in % between the MACCRA and the observations (MACCRA – observation; left of second row), difference between the CIRA and the observation (CIRA – observation; right of second row), difference between the CAMSRA and the observation (CAMSRA – observation; left of third row), and the difference between the control run and the observation (control – observation; right of third row). The data are averaged to 5°×5° (latitude × longitude) and to the altitude bin of 0-3 km.

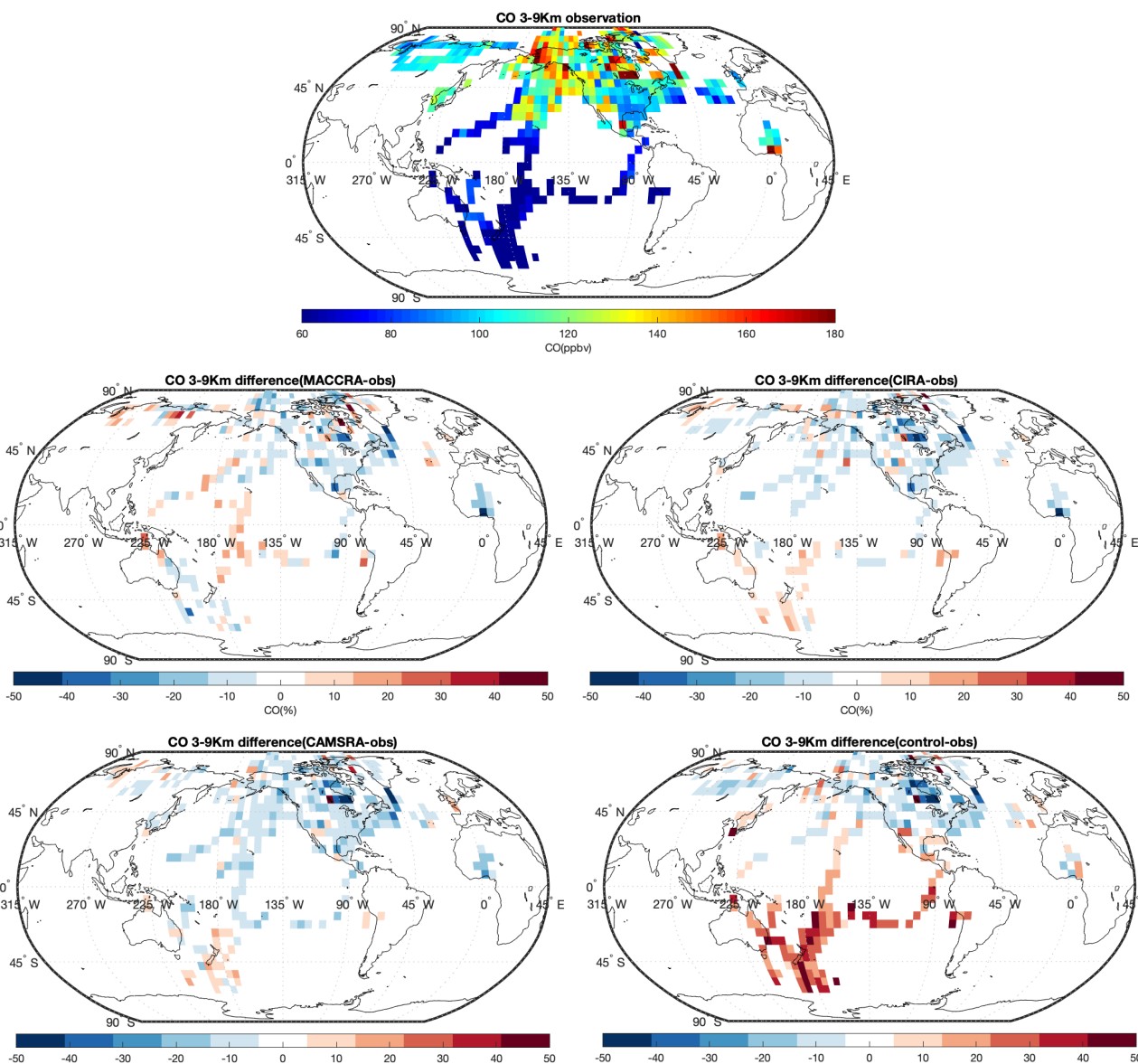

**Figure 6. Same as Figure 5, but for altitude bin of 3-9 km.**


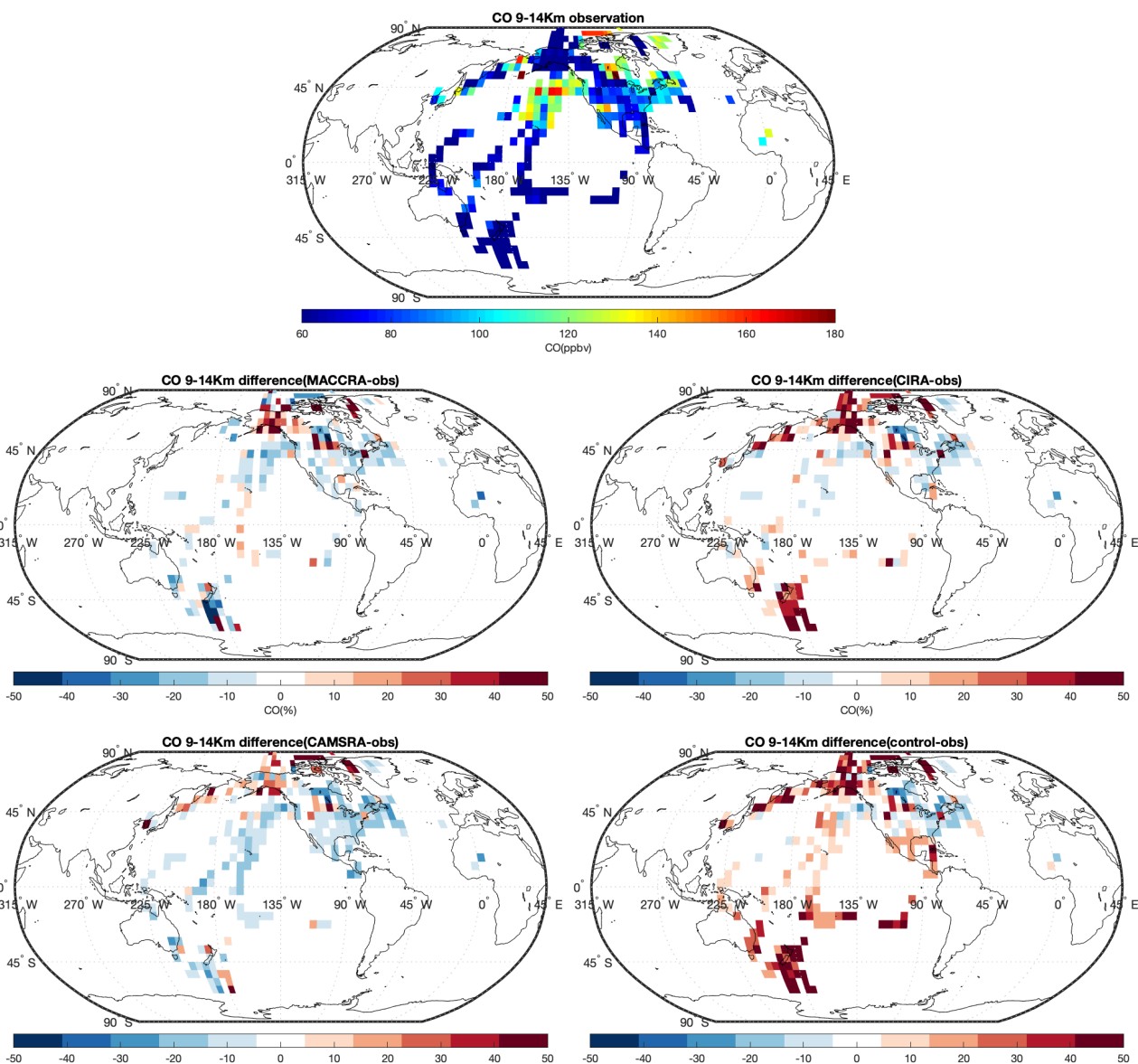

**Figure 7. Same as Figure 5, but for latitude bin of 9-14 km.**


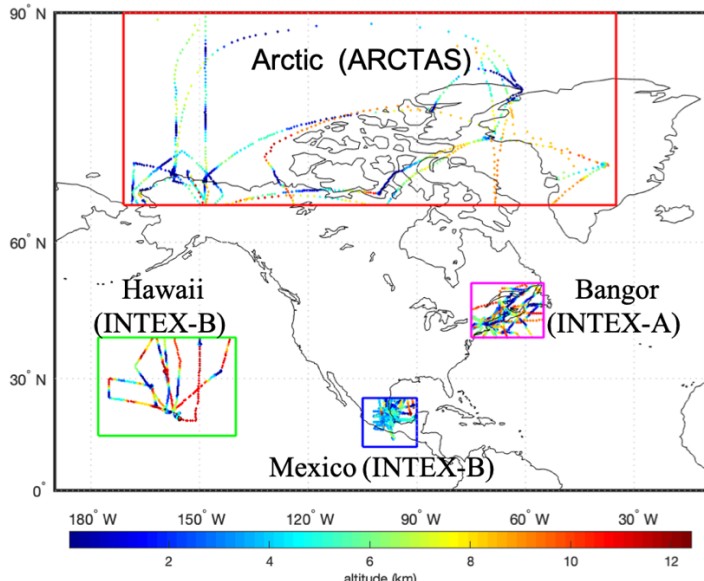

**Figure 8. The location of the four selected regions. The red, green, blue, and magenta rectangles show the Arctic (ARCTAS, 2008.04-07), Hawaii (INTEX-B, 2006.03-05), Mexico (INTEX-B, 2006.03-05), and Bangor (INTEX-A, 2004.07-08), respectively.**


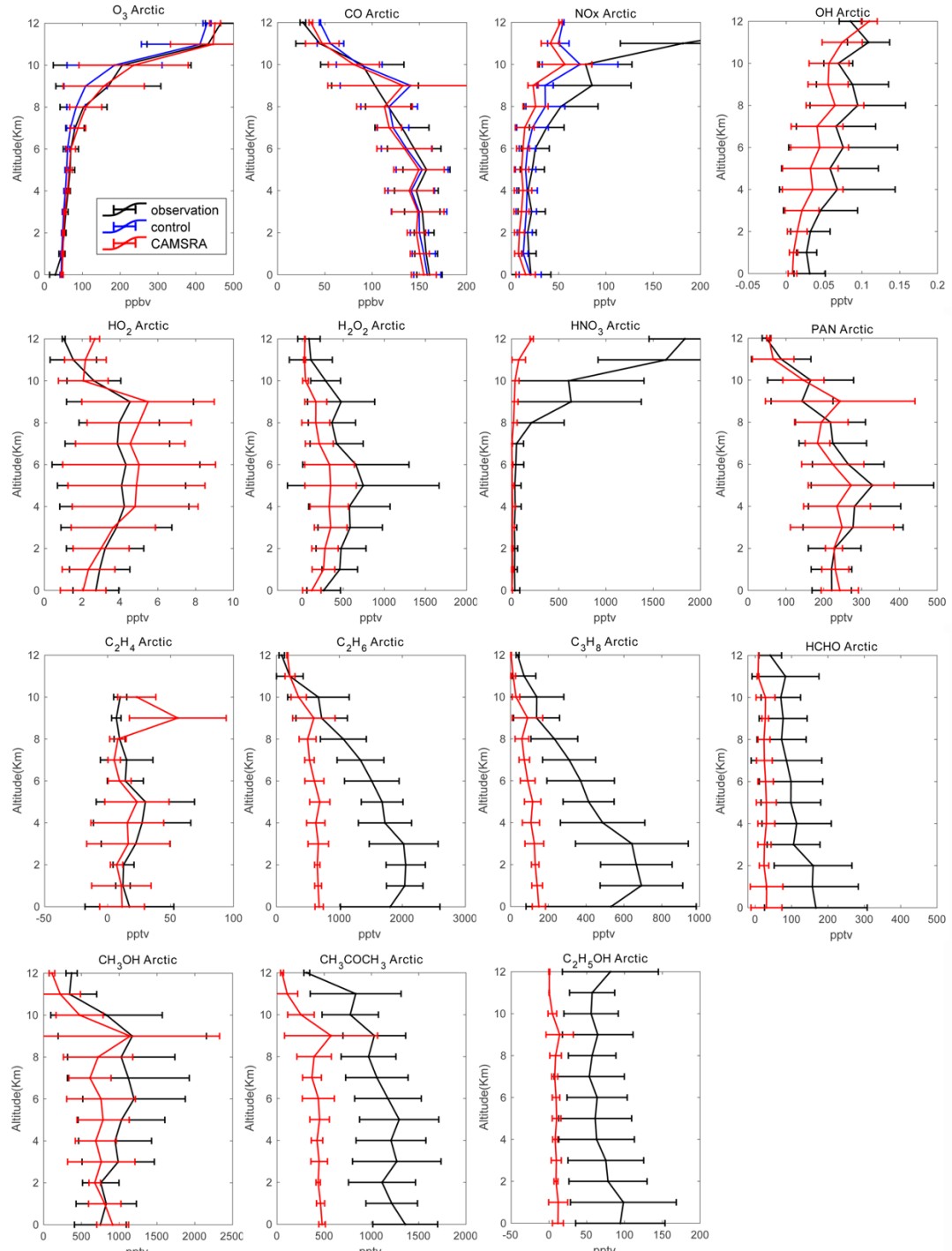

**Figure 9. Averaged profiles of the trace constituents over the Arctic during ARCTAS campaign from April to July, 2008. The black lines are the observations, the red lines correspond to the CAMSRA reanalysis, and the blue lines are the control run (only shown for $O_3$, CO and $NO_X$). The error bars represent the standard deviation of the data/model.**

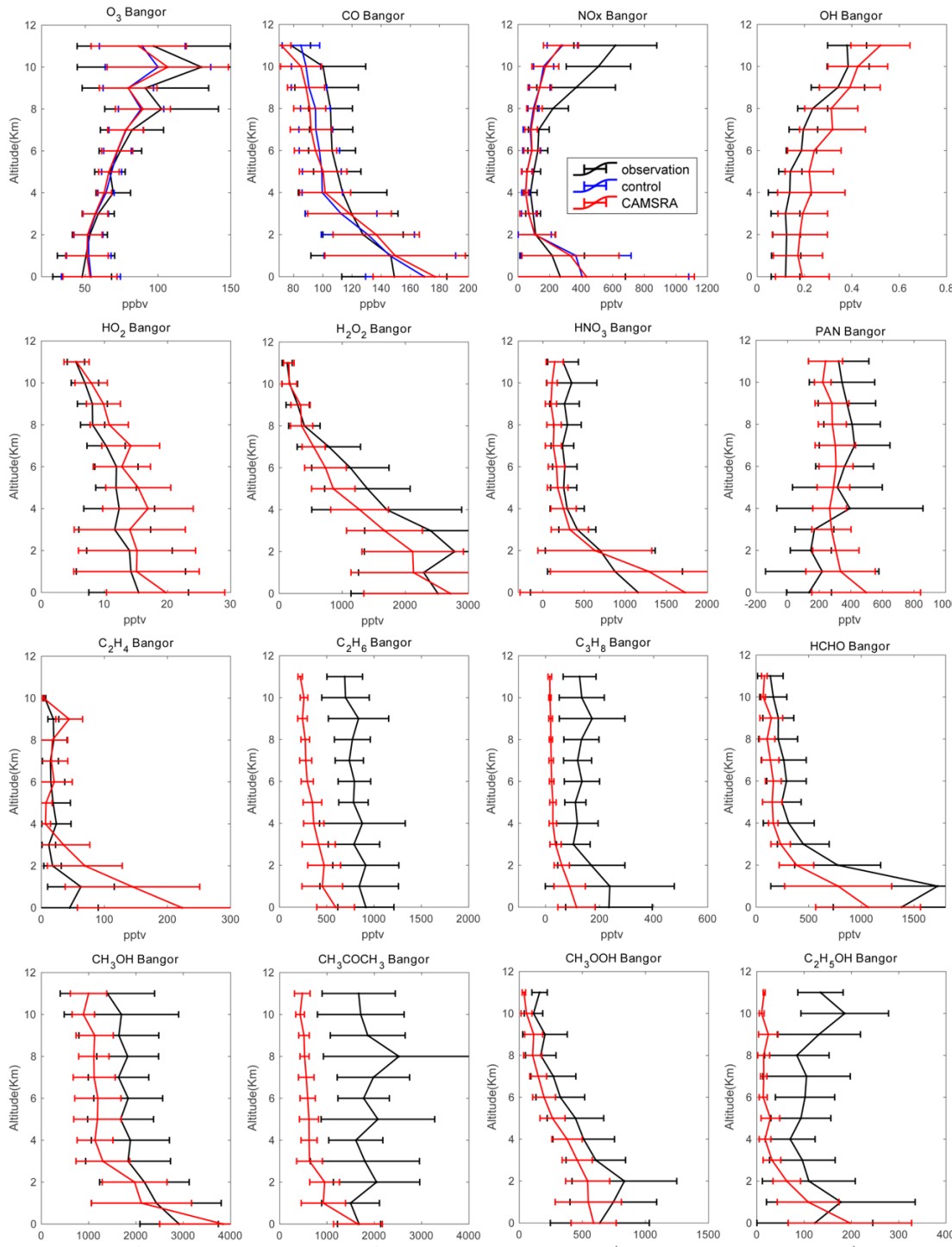

**Figure 10. Averaged profiles of the trace constituents over Bangor during INTEX-A campaign from July to August, 2004. The black lines are the observations, the red lines correspond to the CAMSRA reanalysis, and the blue lines are the control run (only shown for $O_3$, CO and $NO_X$). The error bars represent the standard deviation of the data/model.**

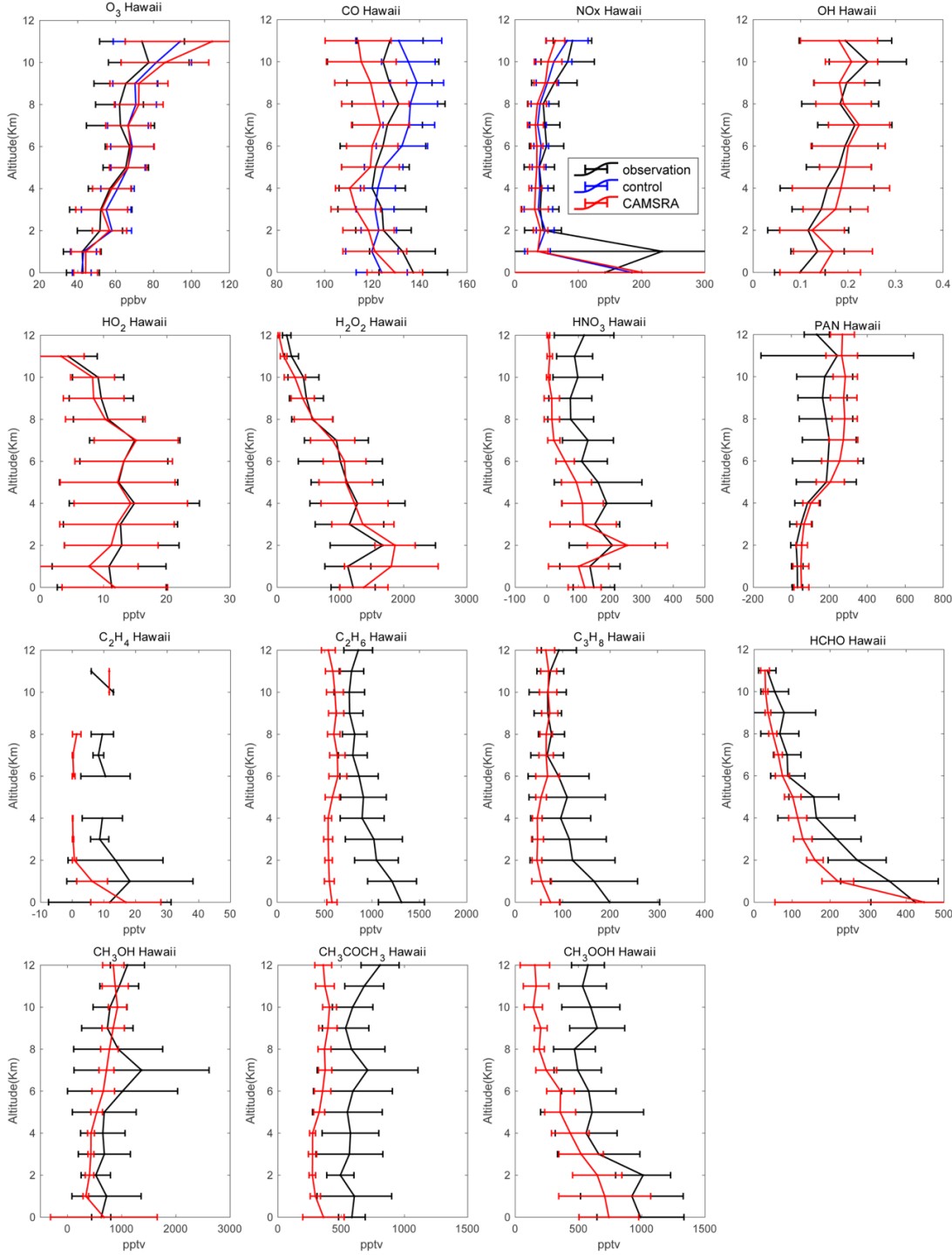

**Figure 11. Averaged profiles of the trace constituents over Hawaii during INTEX-B campaign from March to May, 2006. The black lines are the observations, the red lines correspond to the CAMSRA reanalysis, and the blue lines are the control run (only shown for O₃, CO and NOₓ). The error bars represent the standard deviation of the data/model.**

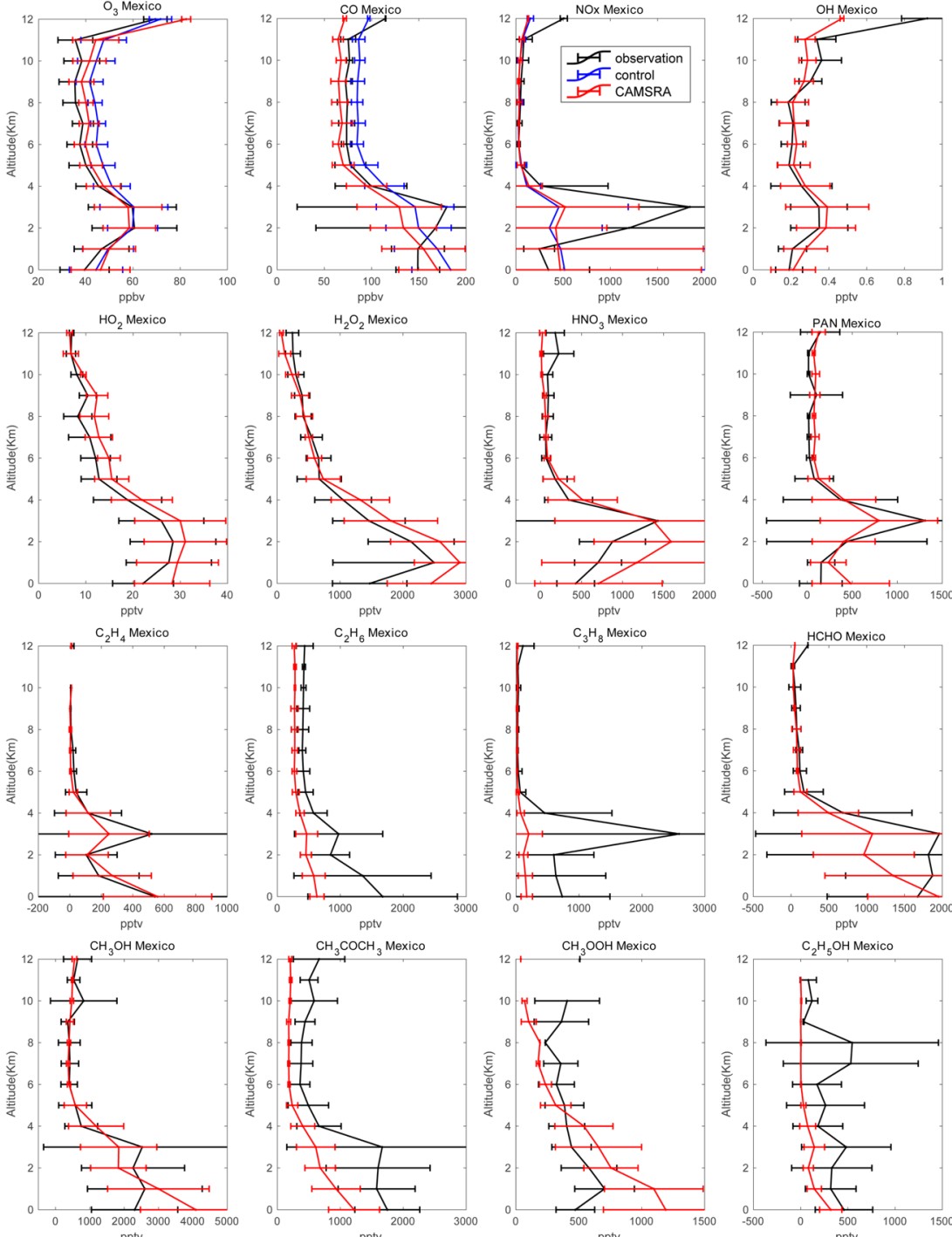

**Figure 12. Averaged profiles of the trace constituents over Mexico during INTEX-B campaign from March to May, 2006. The black lines are the observations, the red lines correspond to the CAMSRA reanalysis, and the blue lines are the control run (only shown for $O_3$, CO and $NO_X$). The error bars represent the standard deviation of the data/model.**


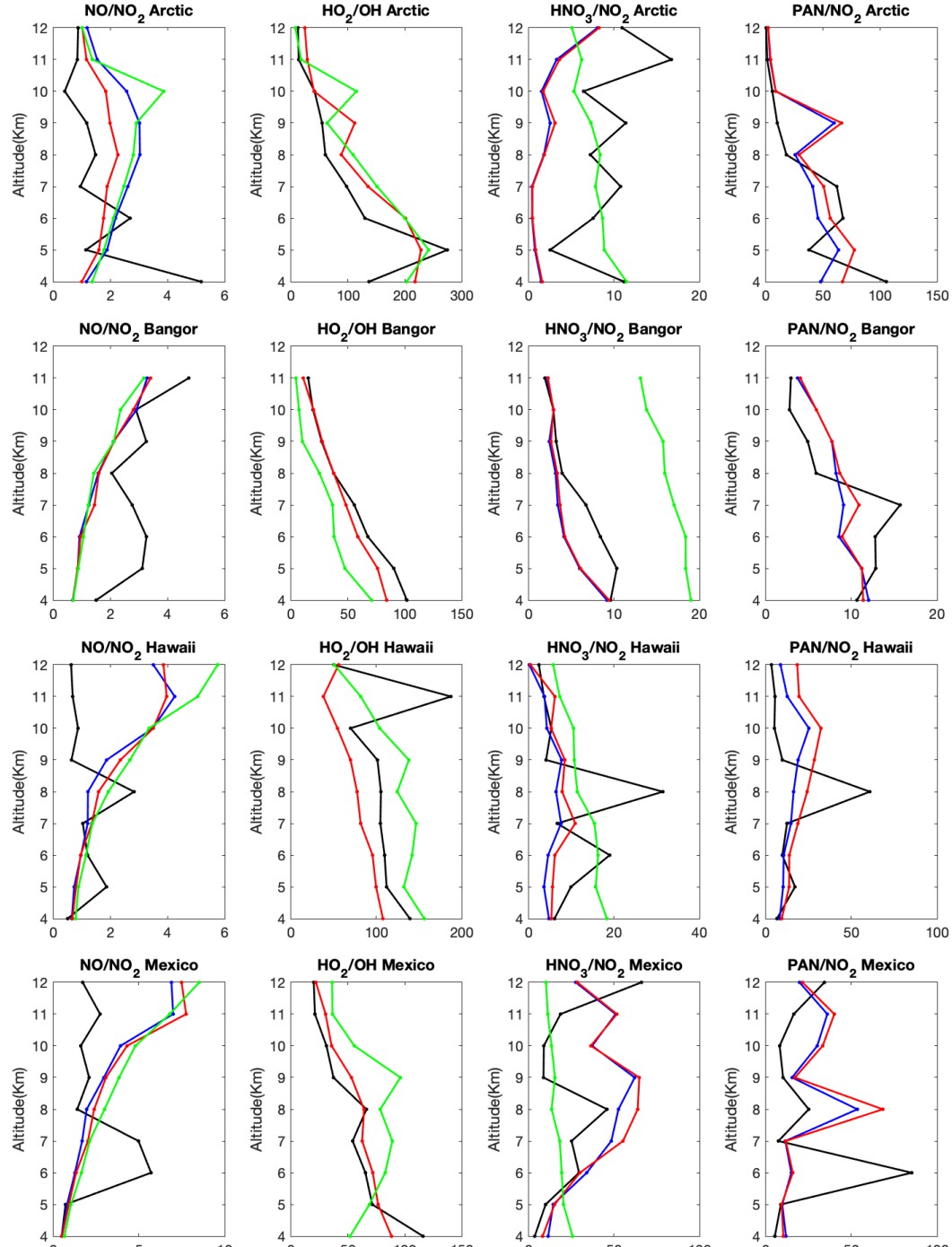

**Figure 13. Concentration ratios of NO/NO₂, HO₂/OH, HNO₃/NO₂ and PAN/NO₂ derived from aircraft measurements (black curves),**
**control runs (blue curves), reanalysis (red curves) and from simple equilibrium relations (except in the case of PAN/NO₂ ratio; green curves). The values are shown in the Arctic, at Bangor, Hawaii and Mexico. Note that, in most cases, the blue and red curves cannot be distinguished.**

# Appendix

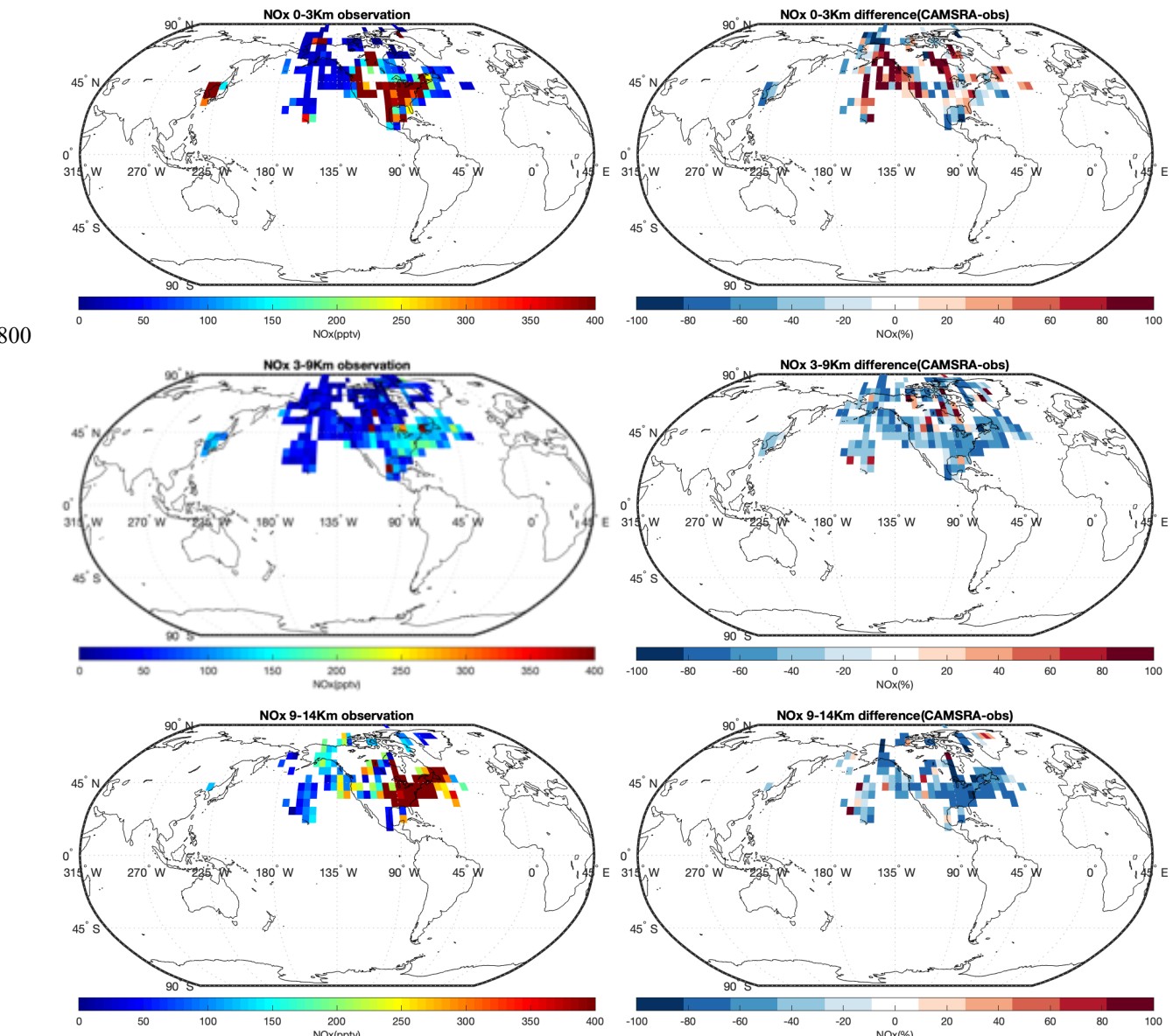


**Figure A1. Campaign observations of NO$_X$ (left panel) and the difference between the CAMSRA and the observations (difference = CAMSRA – observation; right panel). The data are averaged to 5°×5° (latitude × longitude) and to three altitude bins: 0-3 km, 3-9 km, and 9-14 km.**


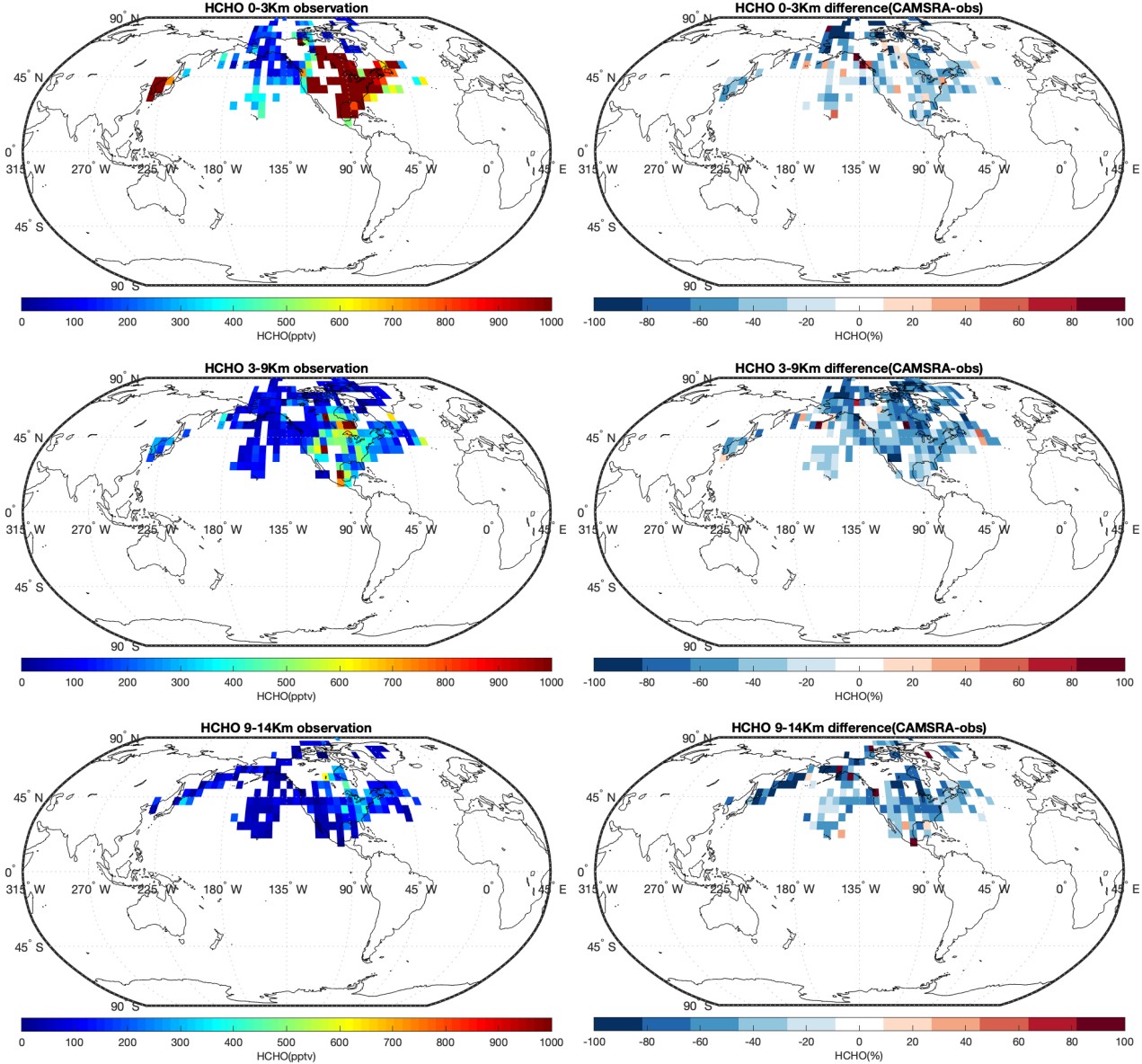


**Figure A2. Same as Figure A1, but for HCHO.**

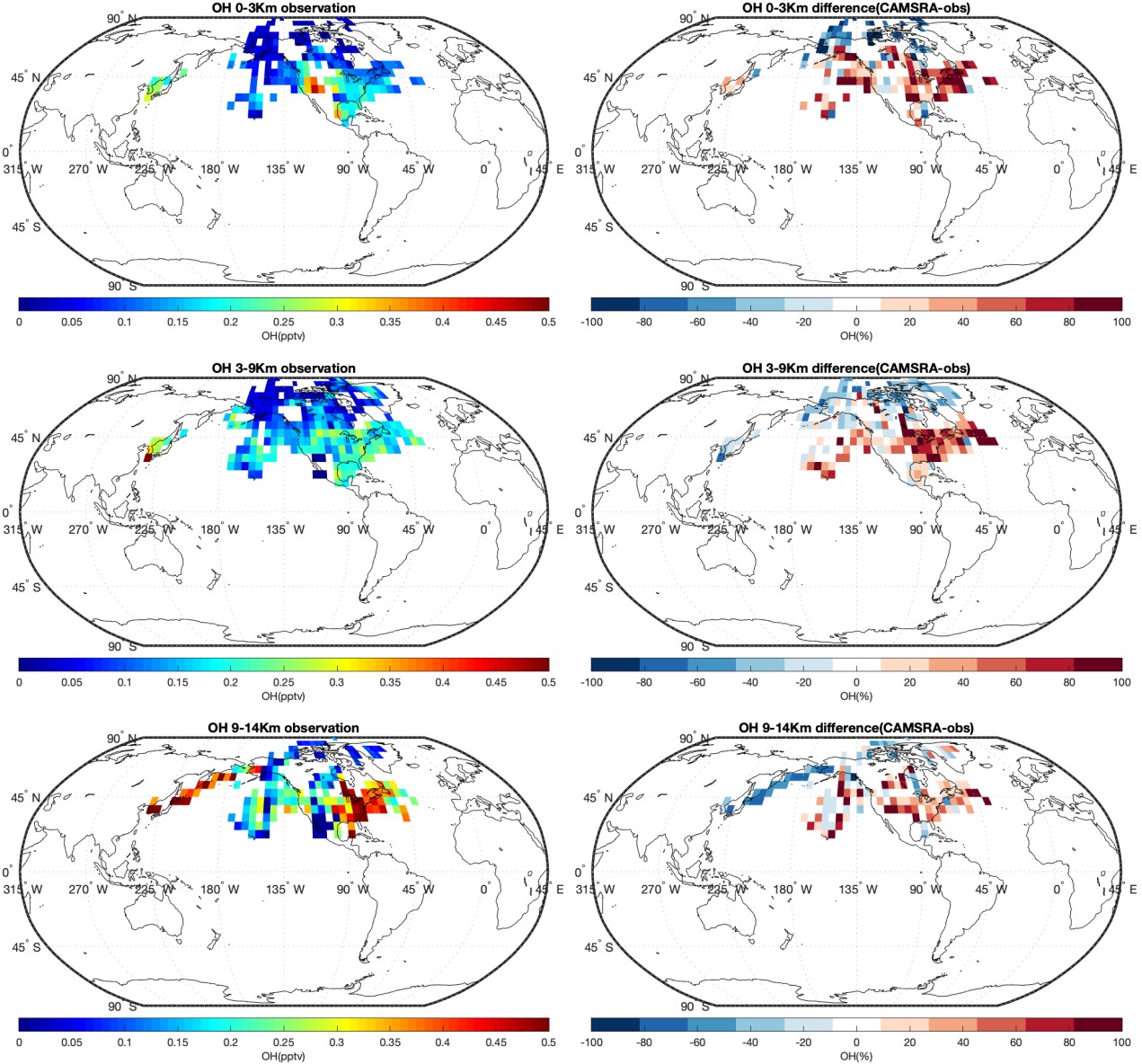

**Figure A3. Same as Figure A1, but for OH.**

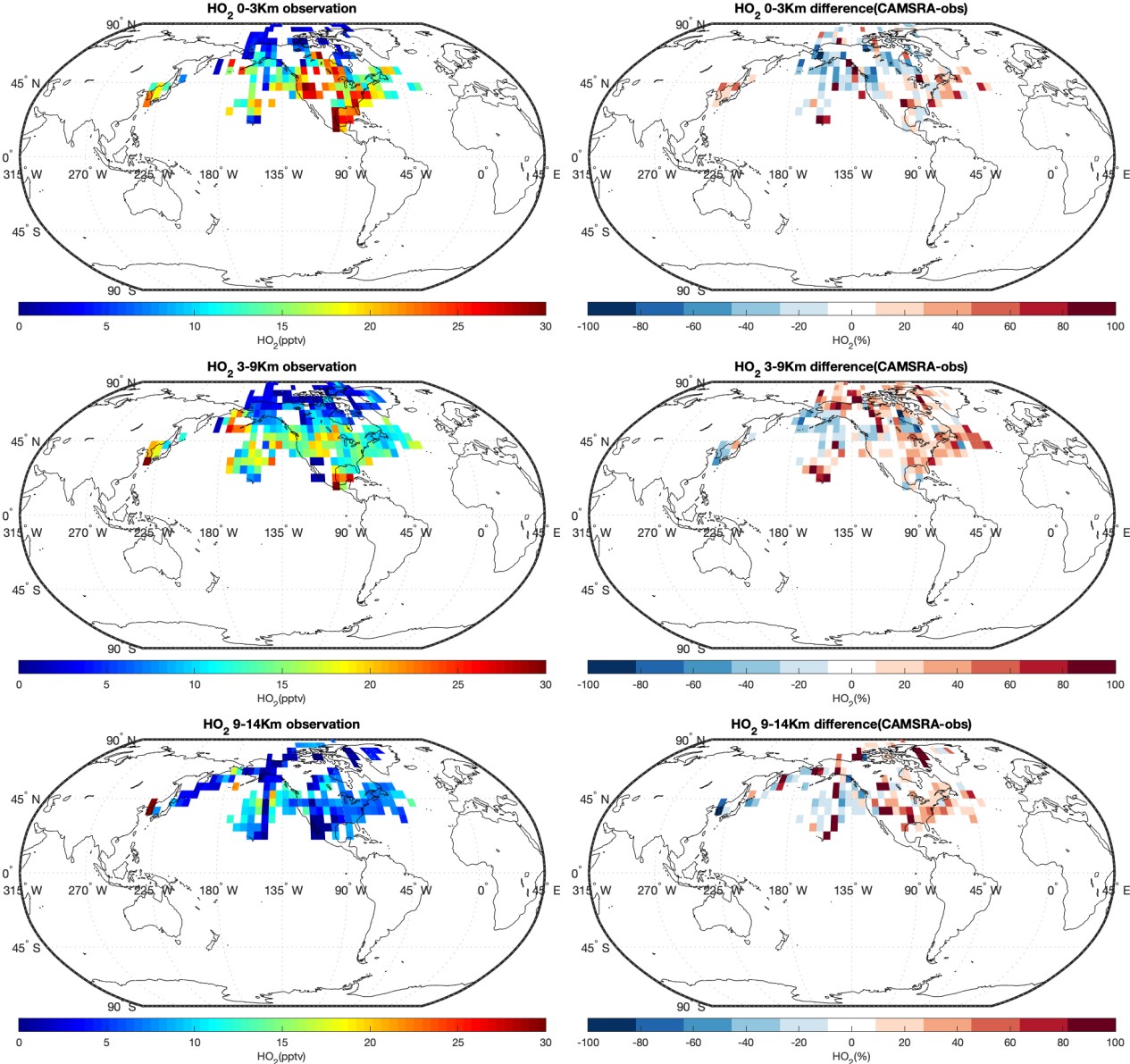

**Figure A4. Same as Figure A1, but for HO$_2$.**