# Peer review of "Evaluation of the CAMS global atmospheric trace gas reanalysis 2003-2016 using aircraft campaign observations"

_Atmospheric Chemistry and Physics, 2019_

## Referee Comment (RC1) · Anonymous Referee #1 · 5 Nov 2019

Wang et al. perform a comprehensive comparison of aircraft observations against three ECMWF reanalyses of atmospheric composition. Their analysis focuses on ozone (O3) and carbon monoxide (CO), but also includes comparisons of other chemical species such as nitrogen oxides, volatile organic compounds (VOCs), and the hydroxyl radical (OH).

Observations from aircraft campaigns constitute a unique resource to evaluate composition models such as the ECMWF reanalysis suite, and the work by Wang et al. offers a meaningful contribution in that regard. Unfortunately, the manuscript contains little interpretation of the results. Rather, it mostly describes the differences between model

and observations, as already shown in the figures. As it stands, it is unclear what the additional insights are compared to e.g. the study by Inness et al. (2019). I recommend adding some high-level discussion to the manuscript in order to explain the results and provide some context. For example, the model seems to generally overpredict OH, which is consistent with an underprediction of CO in the northern hemisphere. Do the authors have an idea why this is the case? Also, model NOx and HNO3 generally seem to be underpredicted in the free troposphere relative to observations, while PAN tends to be higher. Does this suggest that the PAN production rate is too high?

While an in-depth interpretation of all results is out of the scope of this work, highlighting and interpreting some of the main findings from the model-aircraft comparison would go a long way toward making the paper more relevant.

Specific comments:

- Page 2, line 46: the reference for Wagner et al. 2019 is missing in the References section.

- Page 2, line 59: the authors say that the analysis fields for ozone are 'strongly forced by observations', which seems a bit of an odd statement for tropospheric ozone where the constraints provided by the satellites are relatively weak. It would be helpful to expand in a bit more detail how the assimilation impacts tropospheric CO (where the impacts are strong), ozone (some impact), and NO2 (little impact due to the short lifetime).

- Page 4, line 96: the authors use an impressive number of aircraft campaigns for model evaluation. This raises the question how comparable these measurements are? The uncertainties arising from 'mixing and matching' different instruments should be discussed.

- Page 5, line 153: for both the spatial maps and the vertical profiles the authors solely show the mean values. These are useful but don't tell the full story, especially for the

here discussed species that exhibit strong temporal (e.g., daily and seasonal) variations. The ability of the model to capture these variabilities is a key performance metric. It can be somewhat deduced from Tables 4 and 5 but should be discussed in more detail in the manuscript. For example, CAMSRA overestimates ozone in the tropics and Arctic by about 30%. Is this a consistent feature or a seasonal effect (i.e., is the overestimation mostly during spring time –when ARCTAS took place)?

- Page 7, line 215: as part of the CO discussion it would be helpful to discuss the treatment of methane CH4 in the various models. Is CH4 prescribed by all models (and to the same value?), or is it a dynamical species with obvious impacts then on OH and thus CO?

- Page 8, line 247: the assimilation of CO seems to degrade the mean bias relative to the aircraft observations, and generally provide little improvement on the other metrics as well. Do the authors have an explanation for this?

- Page 9, line 271: the authors should add a legend to each figure of vertical profiles to make it easier to distinguish between observations, model, and model background. It would also be helpful to show the observed concentration variation at each level, e.g. by showing the standard deviation (or 25%/75% percentiles) of both the observations and the model comparisons.

---

## Referee Comment (RC2) · Anonymous Referee #2 · 19 Nov 2019

General comments:

This paper presents an evaluation of several CAMS reanalysis products against a suite of aircraft campaign data for multiple chemical species. Evaluation of reanalyses against independent data is an important activity and the evaluation here is rigorous. While the manuscript does a nice job of evaluating the performance of the CAMS re-analyses, there is only limited explanation or analysis of the causes of mismatches between CAMS and observations, or the reasons why the newer reanalysis out-performs the earlier reanalyses in some regions. An interesting result of this paper is that for some species and regions of the atmosphere, the reanalysis has only a little improvement, or even weaker performance, than the control simulation. It would be helpful to have more analysis of why this is the case. Overall, the paper would be strengthened by a more detailed exploration of the underlying causes of the biases, as this could provide guidance for future improvements and provide greater scientific insight.

Specific comments:

Line 56: What other species are assimilated?

Line 57: Which specific satellite observations are assimilated, and how does the assimilation system account for the vertical sensitivities of different satellites? The coarse vertical resolution of satellite data compared to aircraft campaign data is a likely cause of some of the biases against aircraft observations, so this should be discussed in some detail.

Line 88: What does "consistent in time resolution" mean?

Line 186: Please define "good". Some rather large biases are mentioned later in the paragraph.

Lines 210-212: Is that difference in r2 statistically significant? Also, wouldn't we expect a larger improvement since ozone is being assimilated? Is the limited improvement due to limitations or uncertainties in the observations, or something else?

Line 264: Similar to the comment above, why does the NO2 agreement not improve when NO2 is assimilated?

Line 282: Are the differences small because the species are well buffered against changes in O3, NOx, etc., or because the assimilation doesn't change the O3 and NOx concentrations very much?

Line 302: The larger biases of CAMSRA with altitude seems like a surprising result since satellite observations of ozone are available in the stratosphere and upper troposphere but not the boundary layer. It would be nice to relate the discussion of the

vertical profiles to a description of where observations are available to constrain the reanalysis.

Line 358: What emission inventory is used? Does it have known biases, or is this a new finding?

Summary: Can you end with some directions for future improvements and/or a take-home message for the atmospheric chemistry community?

Editorial comments:

Lines 32-33: It seems, then, that the reanalysis covers the period 2003-3018.

Line 52: misplaced comma

Lines 308-309: confusing sentence, please reword

Figs. 9-12: Please use thicker lines so they are easier to see.

---

## Author Comment (AC1) · 15 Jan 2020

We thank the referee#2 for taking the time to read the manuscript and offer helpful comments and suggestions. The referee's comment is repeated with our response in bold. Responses to those comments are listed below: 1. This paper presents an evaluation of several CAMS reanalysis products against a suite of aircraft campaign data for multiple chemical species. Evaluation of reanalyses against independent data is an important activity and the evaluation here is rigorous. While the manuscript does a nice job of evaluating the performance of the CAMS reanalyses, there is only limited explanation or analysis of the causes of mismatches between CAMS and observa-

tions, or the reasons why the newer reanalysis outperforms the earlier reanalyses in some regions. An interesting result of this paper is that for some species and regions of the atmosphere, the reanalysis has only a little improvement, or even weaker performance, than the control simulation. It would be helpful to have more analysis of why this is the case. Overall, the paper would be strengthened by a more detailed exploration of the underlying causes of the biases, as this could provide guidance for future improvements and provide greater scientific insight. Response: We agree with the referee that including further interpretation on the results will give more insight on the model development. We have added more explanations to improve the manuscript as the referee recommended, and in particular we added in the paper a new Section of the concentration ratios of chemically interacting species, so that we can check to what extend the photochemical theory is verified. However, the main point of this paper is to include additional measurements to the routine ones that used by Inness et al. (2019) and Wagner et al. (2019) for the evaluation of the new CAMS reanalysis. The aircraft campaigns provide simultaneous profile measurements of many species such as OH and HO2, which is not within the routine measurements. 2. Line 56: What other species are assimilated? Response: The species that are assimilated are listed in Table 2. 3. Line 57: Which specific satellite observations are assimilated, and how does the assimilation system account for the vertical sensitivities of different satellites? The coarse vertical resolution of satellite data compared to aircraft campaign data is a likely cause of some of the biases against aircraft observations, so this should be discussed in some detail. Response: The satellite used in the assimilations are listed in Table 2. More details for the assimilation have been added in the model description section. 4. Line 88: What does "consistent in time resolution" mean? Response: As shown in Table 1, the time resolution for the analysis fields is six hours for MACCRA and CIRA and three hours for CAMSRA. For the forecast fields, the time resolution is three hours for all the reanalysis versions. So, we used the forecast fields for consistency in the time resolution between the 3 reanalyses. This sentence has been added in the manuscript to make it clear. 5. Line 186: Please define "good". Some rather large biases are

mentioned later in the paragraph. Response: We agree that this expression is not accurate, so we deleted this sentence. 6. Lines 210-212: Is that difference in r2 statistically significant? Also, wouldn't we expect a larger improvement since ozone is being assimilated? Is the limited improvement due to limitations or uncertainties in the observations, or something else? Response: All comparisons show a statistically significant relation between observations and simulations (p<0.001). Therefore, although the difference in r2 only in the range of 0.01-0.05, the results are statistically significant. The simulation is improved to a certain degree with assimilation, for example, r2 and slope increase about 0.05 and 0.1 respectively and RMSE decreases above 4 ppb for O3 after assimilation. However, in the case of tropospheric ozone, the influence of the assimilation is weaker than in the case of stratospheric ozone. 7. Line 264: Similar to the comment above, why does the NO2 agreement not improve when NO2 is assimilated? Response: The impact of the assimilation of tropospheric NO2 column retrievals is small because of the short lifetime of NO2. Even though the assimilation lead to large analysis increments this information was not retained by the model, and most of the impact of the data assimilation was lost from one analysis cycle to the next (Inness et al., 2015). We added this explanation to the manuscript. 8. Line 282: Are the differences small because the species are well buffered against changes in O3, NOx, etc., or because the assimilation doesn't change the O3 and NOx concentrations very much? Response: We would think both are the possible reasons. To further analyze the impact of the assimilations of ozone, CO and NO on the other species, more experiment needs to be run, which is out of the scope of this paper. 9. Line 302: The larger biases of CAMSRA with altitude seems like a surprising result since satellite observations of ozone are available in the stratosphere and upper troposphere but not the boundary layer. It would be nice to relate the discussion of the vertical profiles to a description of where observations are available to constrain the reanalysis. Response: This feature is only shown at Hawaii and Mexico, but not the case at Bangor and the Arctic, so we cannot make the universal conclusion that the biases of CAMSRA increase with altitude. The results show that the model's performance varies from region to region.

The information of the assimilated satellite for different layers are listed in Table 2. 10. Line 358: What emission inventory is used? Does it have known biases, or is this a new finding? Response: The emission inventories are listed in Table 1. Their choice has been made by the CAMS project at ECMWF based on what was estimated as the best inventories available at that time. For example, the anthropogenic emissions are based on the MacCity inventory. No systematic bias is known for these inventories, but uncertainties exist. Emission inventories are constantly updated to address uncertainties and to account for changes in emissions. 11. Summary: Can you end with some directions for future improvements and/or a take-home message for the atmospheric chemistry community? Response: We added a sentence in the conclusions calling for an improvement in the adopted surface emissions of hydrocarbons and on the need to assimilate, if possible, some organic species in addition to formaldehyde. 12. Lines 32-33: It seems, then, that the reanalysis covers the period 2003-3018. Response: Yes, the reanalysis now covers 2003-2018, but we only evaluate the first release of the reanalysis, which is 2003-2016. 13. Line 52: misplaced comma Response: Corrected. 14. Lines 308-309: confusing sentence, please reword Response: Corrected. 15. Figs. 9-12: Please use thicker lines so they are easier to see. Response: We have modified the figures as suggested.

---

## Author Comment (AC2) · 15 Jan 2020

We thank the referee#1 for taking the time to read the manuscript and offer helpful comments and suggestions. The referee's comment is repeated with our response in bold. Responses to those comments are listed below: 1. Wang et al. perform a comprehensive comparison of aircraft observations against three ECMWF reanalyses of atmospheric composition. Their analysis focuses on ozone (O3) and carbon monoxide (CO), but also includes comparisons of other chemical species such as nitrogen oxides, volatile organic compounds (VOCs), and the hydroxyl radical (OH). Observations from aircraft campaigns constitute a unique resource to evaluate composition models

such as the ECMWF reanalysis suite, and the work by Wang et al. offers a meaningful contribution in that regard. Unfortunately, the manuscript contains little interpretation of the results. Rather, it mostly describes the differences between model and observations, as already shown in the figures. As it stands, it is unclear what the additional insights are compared to e.g. the study by Inness et al. (2019). I recommend adding some high-level discussion to the manuscript in order to explain the results and provide some context. For example, the model seems to generally overpredict OH, which is consistent with an underprediction of CO in the northern hemisphere. Do the authors have an idea why this is the case? Also, model NOx and HNO3 generally seem to be underpredicted in the free troposphere relative to observations, while PAN tends to be higher. Does this suggest that the PAN production rate is too high? While an in-depth interpretation of all results is out of the scope of this work, highlighting and interpreting some of the main findings from the model-aircraft comparison would go a long way toward making the paper more relevant. Response: We agree with the referee that including further interpretation on the results will give more insight on the model development. We have added more explanations to improve the manuscript as the referee recommended, and in particular we added in the paper a new Section of the concentration ratios of chemically interacting species, so that we can check to what extend the photochemical theory is verified. Note, however, that the main point of this paper is to include additional measurements to the routine observations used by Inness et al. (2019) and Wagner et al. (2019) for the evaluation of the new CAMS reanalysis. The aircraft campaigns provide simultaneous profile measurements of many species such as OH and HO2, which are not taken into consideration in the routine evaluation. Page 2, line 46: the reference for Wagner et al. 2019 is missing in the References section. Response: Wagner et al. 2019 has been updated in the reference section. 2. Page 2, line 59: the authors say that the analysis fields for ozone are 'strongly forced by observations', which seems a bit of an odd statement for tropospheric ozone where the constraints provided by the satellites are relatively weak. It would be helpful to expand in a bit more detail how the assimilation impacts tropospheric CO (where

the impacts are strong), ozone (some impact), and NO2 (little impact due to the short lifetime). Response: We modified the sentence and added more explanations in the model description section. 3. Page 4, line 96: the authors use an impressive number of aircraft campaigns for model evaluation. This raises the question how comparable these measurements are? The uncertainties arising from 'mixing and matching' different instruments should be discussed. Response: Although the campaigns used different instruments to measure O3 and CO, the instruments were all calibrated and have relatively small uncertainties in a range of 3-5 ppb for O3 and 2-5 ppb for CO. In some campaigns that measured the same species using several instruments, we compared the simultaneous measurements, and they are in good agreement. We averaged the data to further reduce the uncertainties. We have added one paragraph to address this issue in the campaign description section. 4. Page 5, line 153: for both the spatial maps and the vertical profiles the authors solely show the mean values. These are useful but don't tell the full story, especially for the here discussed species that exhibit strong temporal (e.g., daily and seasonal) variations. The ability of the model to capture these variabilities is a key performance metric. It can be somewhat deduced from Tables 4 and 5 but should be discussed in more detail in the manuscript. For example, CAMSRA overestimates ozone in the tropics and Arctic by about 30%. Is this a consistent feature or a seasonal effect (i.e., is the overestimation mostly during spring time –when ARCTAS took place)? Response: The strong spatial variation only imply that the model's performance varies with space. But we cannot be sure that whether it is a seasonal effect or regional feature due to the sparse measurements (e.g., there are no different campaigns performed in the same region in different seasons). 5. Page 7, line 215: as part of the CO discussion it would be helpful to discuss the treatment of methane CH4 in the various models. Is CH4 prescribed by all models (and to the same value?), or is it a dynamical species with obvious impacts then on OH and thus CO? Response: Methane is calculated in the simulation with constant surface concentrations (as opposed to emissions) applied as lower boundary conditions. The destruction of methane results from the presence of OH that is calculated by the

model. OH and thus CO are affected by the calculated methane. We have added a sentence at the beginning of Section 2 to specify how methane is calculated. 6. Page 8, line 247: the assimilation of CO seems to degrade the mean bias relative to the aircraft observations, and generally provide little improvement on the other metrics as well. Do the authors have an explanation for this? Response: The control run largely overestimates the CO concentrations in the Southern Hemisphere as shown in Figure 5 – 7. The assimilation reduces the positive bias in the Southern Hemisphere thus degrade the mean bias in Table 5. This statement is added in the manuscript. For the all data analysis, the calculated numbers are largely affected by the extreme values that the satellites cannot capture because of the coarser resolution than the aircraft measurements. After filtering out the plumes, the CAMSRA has larger r2 (0.71) and slope (0.78) than the control run (0.66 and 0.75). 7. Page 9, line 271: the authors should add a legend to each figure of vertical profiles to make it easier to distinguish between observations, model, and model background. It would also be helpful to show the observed concentration variation at each level, e.g. by showing the standard deviation (or 25%/75% percentiles) of both the observations and the model comparisons. Response: We added the legend and the standard deviation in the plots.